# Association of snR190 snoRNA chaperone with early pre-60S particles is regulated by the RNA helicase Dbp7 in yeast

Mariam Jaafar[1,2,9], Julia Contreras[3,4], Carine Dominique[1], Sara Martín-Villanueva [3], Régine Capeyrou[1], Patrice Vitali[1], Olga Rodríguez-Galán[3,4], Carmen Velasco[3,5], Odile Humbert[1], Nicholas J. Watkins[6], Eduardo Villalobo [3,5], Katherine E. Bohnsack [7], Markus T. Bohnsack [7,8], Yves Henry[1], Raghida Abou Merhi[2], Jesús de la Cruz [3,4] & Anthony K. Henras [1✉]

Synthesis of eukaryotic ribosomes involves the assembly and maturation of precursor particles (pre-ribosomal particles) containing ribosomal RNA (rRNA) precursors, ribosomal proteins (RPs) and a *plethora* of assembly factors (AFs). Formation of the earliest precursors of the 60S ribosomal subunit (pre-60S r-particle) is among the least understood stages of ribosome biogenesis. It involves the Npa1 complex, a protein module suggested to play a key role in the early structuring of the pre-rRNA. Npa1 displays genetic interactions with the DExD-box protein Dbp7 and interacts physically with the snR190 box C/D snoRNA. We show here that snR190 functions as a snoRNA chaperone, which likely cooperates with the Npa1 complex to initiate compaction of the pre-rRNA in early pre-60S r-particles. We further show that Dbp7 regulates the dynamic base-pairing between snR190 and the pre-rRNA within the earliest pre-60S r-particles, thereby participating in structuring the peptidyl transferase center (PTC) of the large ribosomal subunit.

[1] Molecular, Cellular and Developmental Biology Unit (MCD), Centre de Biologie Intégrative (CBI), Université de Toulouse, CNRS, UPS, 31062 Toulouse, France. [2] Genomic Stability and Biotherapy (GSBT) Laboratory, Faculty of Sciences, Rafik Hariri Campus, Lebanese University, Beirut, Lebanon. [3] Instituto de Biomedicina de Sevilla (IBiS), Hospital Universitario Virgen del Rocío/CSIC/Universidad de Sevilla, 41013 Seville, Spain. [4] Departamento de Genética, Facultad de Biología, Universidad de Sevilla, 41012 Seville, Spain. [5] Departamento de Microbiología, Facultad de Biología, Universidad de Sevilla, 41012 Seville, Spain. [6] Institute for Cell and Molecular Biosciences, The Medical School, Newcastle University, Newcastle upon Tyne NE1 7RU, UK. [7] Department of Molecular Biology, University Medical Centre Göttingen, 37073 Göttingen, Germany. [8] Göttingen Center for Molecular Biosciences, Georg-August University Göttingen, 37077 Göttingen, Germany. [9] Present address: Cancer Research Center of Lyon (CRCL), 69 008 Lyon, France. ✉email: anthony.henras@univ-tlse3.fr

Ribosomes are universal molecular machines responsible for protein synthesis. The first crystal structures of ribosomal subunits (r-subunits)[1,2] revealed that the ribosome is a ribozyme[3], namely an RNA-based enzyme. This implies that an essential aspect of ribosome biogenesis is the accurate folding of the rRNAs to generate their functional structure. In yeast, rRNAs contained in the 40S (18S) and 60S (5S, 5.8S, and 25S) r-subunits are generated by transcription of rDNA genes by RNA polymerases I and III (Pol I and Pol III). Pol I-mediated transcription generates a primary transcript containing the sequences of 18S, 5.8S, and 25S rRNAs. Maturation of this pre-rRNA occurs within a series of preribosomal particles containing also RPs and numerous AFs. This process comprises pre-rRNA nucleoside modifications, cleavage and folding events and the incorporation and proper positioning of RPs that gradually lead to the formation of mature r-subunits[4–6].

Assembly of the earliest precursors to the large (60S) r-subunit is the least understood step of eukaryotic ribosome biogenesis. These particles contain numerous small nucleolar ribonucleoprotein particles (snoRNPs)[7], which belong to the C/D- and H/ACA-type families. C/D-type snoRNPs typically catalyze the 2'-O-methylation of selected nucleosides. They contain a box C/D small nucleolar RNA (snoRNA) associated with a protein core including the methyltransferase Nop1. H/ACA-type snoRNPs are composed of a box H/ACA snoRNA and several proteins including the pseudouridine synthase Cbf5, which isomerizes uridines into pseudouridines. In both cases, the snoRNAs contain antisense elements that base-pair to target sequences on the pre-rRNA, and function as a guide for specific nucleoside modifications[8,9]. A subset of snoRNAs from both families do not function as modification guides but as RNA chaperones, assisting folding of the pre-rRNA to promote proper processing. In yeast, four snoRNP chaperones, containing U3, U14, snR30, and snR10 snoRNAs, are involved in the biogenesis of the small r-subunit[10–13]. An important issue that remains to be addressed is when and how snoRNA base-pairings with the pre-rRNA are disrupted, given that some of these interactions are thermodynamically highly stable. NTP-dependent RNA helicases have the potential to dissociate snoRNA-pre-rRNA interactions. Twelve RNA helicases (Dbp2, Dbp3, Dbp6, Dbp7, Dbp9, Dbp10, Drs1, Mak5, Mtr4, Spb4, Prp43, and Has1) participate in the synthesis of the large r-subunit[14,15] and interestingly, eight of them (Dbp3, Dbp6, Dbp7, Dbp9, Drs1, Has1, Mak5, and Prp43) are present in early pre-60S r-particles[7]. These proteins are believed to fulfill various functions in the remodeling events occurring within pre-60S r-particles, such as promoting formation, stability or dissociation of RNA secondary structures or regulating the binding or dissociation of proteins[16,17]. Among these enzymes, only the RNA helicase Prp43 and Dbp3 have been shown to function in the release of snoRNAs[18,19].

The structural organization of the RNA components of the earliest pre-60S r-particles remains ill-defined. In the mature 60S r-subunit, the 25S and 5.8S rRNAs are organized in six structural domains (I to VI), each beginning with short stems referred to as root helices (Supplementary Fig. 1). These root helices are clustered in the mature particles and their clustering and compaction has been proposed to occur early during assembly of pre-60S r-particles[20]. In the earliest pre-60S r-particles, root helix I and base-pairing interactions between the 25S and 5.8S rRNAs, key determinants of domain I folding, are already established[21]. In the earliest pre-60S particle cryo-electron microscopy structure reported to date (State A[22]), the root helix of domain I is formed, and domains I, II and part of domain VI have acquired their final secondary structure. These structured domains form a rigid platform, corresponding to the solvent-exposed side of the mature subunit, onto which takes place assembly and compaction

of the other domains[22]. Failure to correctly complete these early folding events is believed to lead to turnover of pre-60S r-particles.

Formation and/or stability of the earliest pre-60S r-particles in yeast require the Npa1 complex, a protein module composed of 5 AFs: Npa1, Npa2, Nop8, Rsa3, and the RNA helicase Dbp6[7,23,24]. Mapping of RNA interaction sites of Npa1 using the crosslinking and analysis of cDNA approach (CRAC) revealed that Npa1 interacts with both the 5' (domain I) and 3' (domains V and VI) regions of the 25S rRNA (Supplementary Fig. 1), and has therefore the potential to act as a long-range, physical link between domains I, V, and VI[25]. The Npa1 complex may therefore promote or stabilize circularization of the 25S rRNA in the earliest stages of pre-60S r-particle assembly, by tethering the 5' and 3' domains. Npa1 also crosslinks to a subset of C/D- and H/ACA-type snoRNAs involved in the chemical modification of nucleosides of the decoding center and PTC of the large r-subunit. Interestingly, the snR190 box C/D snoRNA is particularly efficiently crosslinked to Npa1. Its antisense sequence upstream of box D' (box A, Supplementary Fig. 2a) is complementary to a sequence embedded within helix H73 in domain V of the 25S rRNA and it is predicted to guide ribose 2'-O-methylation at position G2395 (Supplementary Fig. 1). However, this methylation has not been detected in recent large scale studies[26–28]. snR190 also contains another 15 nucleoside sequence (box B, Supplementary Fig. 2a) that is perfectly complementary to a sequence in domain I of 25S rRNA near the Npa1 binding site (Supplementary Fig. 1). Like the Npa1 complex, snR190 therefore also has the potential to make a physical link between the 5' and 3' domains of the 25S rRNA.

Here, we provide evidence that yeast snR190 does not function as a methylation guide snoRNP, but rather as a placeholder preventing specific 25S rRNA sequences from establishing inappropriate interactions during pre-rRNA folding. Moreover, our data strongly support the model that Dbp7 regulates the dynamics of base pairing between snR190 and the 25S rRNA within early pre-60S r-particles.

## Results

**Genetic link between Dbp7 and snR190 base-pairing site on 25S rRNA.** The absence of Dbp7 impairs production of the 27S and 7S pre-rRNA intermediates during early pre-60S r-particle maturation, and compromises synthesis of mature 25S and 5.8S rRNAs and the 60S subunit[29,30]. The precise molecular function of Dbp7 in this process remains elusive. *DBP7* displays genetic interactions with most genes encoding members of the Npa1 complex[24], suggesting that Dbp7 function may be related to these AFs[25]. To gain further insight into the function of Dbp7, we exploited the severe growth defect of the *dbp7Δ* strain to isolate suppressors bearing mutations in rRNAs. We made use of the yeast strain NOY891[31], in which all the chromosomal rDNA genes are deleted and the 35S pre-rRNA is expressed from a plasmid (pNOY353, 2 μ, *TRP1*) driven by the galactose-inducible *GAL7* promoter[31]. We further engineered this strain to express a HA-tagged version of Dbp7 under control of another galactose-inducible promoter (*GAL1::HA-DBP7*, strain EMY65). To identify rRNA suppressors of Dbp7 loss-of-function, a mutant library of a second plasmid, (pNOY373, 2 μ, *LEU2*) expressing the 35S pre-rRNA from its natural Pol I-dependent promoter, was transformed into strain EMY65. Transformants were screened on selective SD medium for faster-growing clones, condition where neither pNOY353 nor the *GAL1::HA-DBP7* construct are expressed (Supplementary Fig. 3). Only two plasmids recovered from candidate clones, referred to as Sup. #2 and Sup. #10, reproducibly improved the

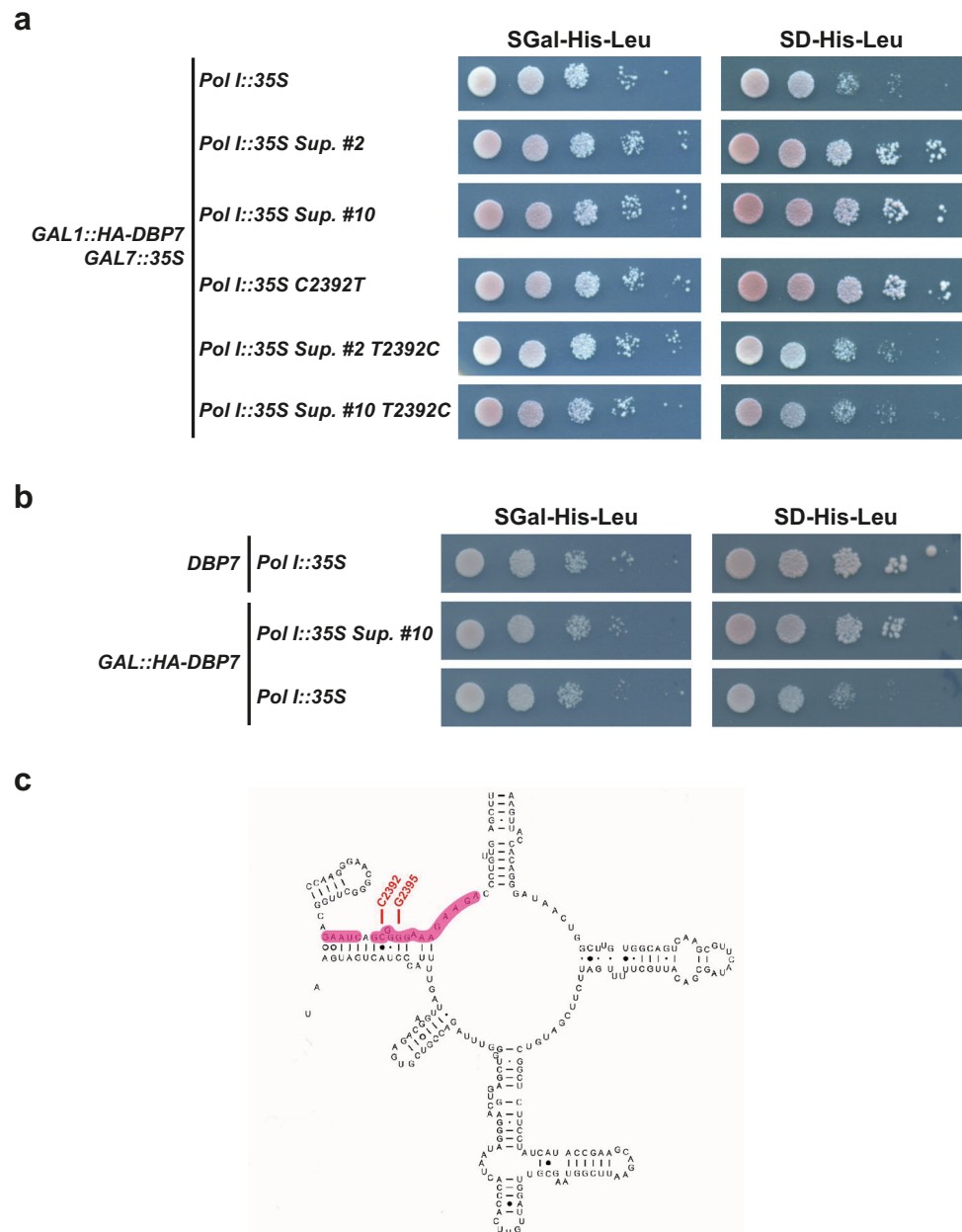

**Fig. 1 rRNA suppressors of the Dbp7 depletion. a** Serial dilution growth test of control (*Pol I::35S*) and suppressor (*Pol I::35S Sup. #2 and Pol I::35S Sup. #10*) strains isolated in the genetic screen for rRNA suppressors of Dbp7 depletion. Growth of strain EMY65 transformed with plasmid bearing the specific C2392T mutation (*Pol I::35S C2392T*), or with plasmids Sup. #2 and Sup. #10 in which mutation T2392 has been reverted to the original C2392 (*Pol I::35S Sup. #2 T2392C* and *Pol I::35S Sup. #10 T2392C*) is also tested. All strains grow similarly on selective SGal medium, which allows expression of Dbp7. On selective SD medium, the control strain shows a strong growth defect resulting from Dbp7 depletion, which is alleviated in all the strains expressing the mutant 25S rRNA with the C2392U substitution. Plates were incubated at 30 °C for 4 days. **b** Serial dilution growth test of strain NOY891 (original Nomura's lab strain, *DBP7*) and strain EMY65 (*GAL::HA-DBP7*) transformed with either the control plasmid (*Pol I::35S*) or the suppressor Sup. #10 plasmid (*Pol I:: 35S Sup. #10*). All strains grow similarly on selective SGal medium. On selective SD medium, strain EMY65 transformed with the control plasmid showed a strong growth defect, which is only partially rescued, with respect to the reference NOY891 strain, by the Sup. #10 plasmid. Plates were incubated at 30 °C for 4 days. **c** Schematic representation of the secondary structure of the 25S rRNA in the PTC region (domain V). The sequence complementary to snR190 box A is highlighted in pink. The predicted methylation site of snR190 (G2395) and the nucleoside found mutated in the genetic screen (C2392U) are indicated in red.

growth of strain EMY65 when retransformed (Fig. 1a). Their rDNA inserts were found to carry different mutations (Supplementary Table 1). Strikingly, both featured, among others, the same rRNA mutation, a C-to-U transition at position C2392 in helix H73 of the 25S rRNA sequence. To confirm that C2392U was itself responsible for the suppression effect, we specifically introduced the C2392T mutation in the wild-type

pNOY373 plasmid. In parallel, we also introduced the T2392C reversion in the Sup. #2 and Sup. #10 plasmids. Growth of the resulting strains at 30 °C on selective SGal and SD media showed that the C2392U substitution was both necessary and sufficient to rescue the growth defect of the *dbp7Δ* strain (Fig. 1a). This suppression effect was not complete, as growth of the EMY65 strain transformed with the Sup. #10 plasmid on SD

medium was slower than that of the original NOY891 strain expressing Dbp7 (Fig. 1b).

Helix H73 corresponds to the root helix of domain V of the 25S rRNA, which forms the PTC of the mature 60S r-subunit (Fig. 1c and Supplementary Fig. 1). Interestingly, the C2392U substitution lies in the immediate vicinity of G2395, predicted to undergo ribose 2'-O-methylation guided by the snR190 box C/D snoRNA (Supplementary Fig. 1), and is expected to weaken the base pairing between snR190 and its complementary sequence on the pre-rRNA. snR190 is the snoRNA most efficiently crosslinked to Npa1 in CRAC experiments[25]. This intimate biochemical relationship between Npa1 and snR190 along with (i) the genetic interaction between Dbp7 and Npa1, and (ii) the suppression of the growth defect of the loss-of-function of Dbp7 by a mutation in the 25S rRNA predicted to weaken the base pairing with snR190, led us to hypothesize that the function of Dbp7 might be linked to that of snR190.

**snR190 is required for optimal 60S r-subunit biogenesis.** In yeast, snR190 is expressed from a dicistronic precursor transcript also supporting expression of U14, an essential box C/D snoRNA that functions as an RNA chaperone in the maturation of the 18S rRNA[32]. This particular context precluded using conventional genomic knockout approaches to inactivate snR190 expression. To interfere with snR190 expression post-transcriptionally, we mutagenized the genomic sequences encoding box C and the terminal stem of the snoRNA (snr190-[mut.C], Supplementary Fig. 2b) using the CRISPR-Cas9 approach[33] (see "Materials and Methods" section for details). We selected two independent snr190-[mut.C] clones in two different yeast genetic backgrounds, W303 and BY4741, and confirmed snR190 depletion by northern blotting (Fig. 2a and Supplementary Fig. 4a). Strains having undergone the mutagenesis procedure, but which lacked mutations in SNR190 were used as controls (referred to as "WT" on the figures).

To assess the impact of the loss-of-function of snR190 on cell fitness, we carried out growth tests in liquid YPD medium at 30 °C. We observed that the snr190-[mut.C] strains showed a moderate but reproducible decrease in growth rate (Supplementary Table 2 and Supplementary Fig. 15). In order to verify that this defect was not due to off-target mutations introduced by the CRISPR-Cas9 approach, we rescued snR190 expression in these strains. The SNR190-U14 gene was PCR amplified from yeast genomic DNA and inserted into a centromeric vector. The resulting plasmid, as well as an empty vector (E.V.), as a negative control, were transformed into the snr190-[mut.C] strains. We confirmed by northern blotting re-expression of snR190 in the snr190-[mut.C] strain in the W303 background (Fig. 2b). Ectopic expression of snR190 in the snr190-[mut.C] strains (BY4741 and W303 backgrounds) fully restored wild-type growth rates (Supplementary Table 2 and Supplementary Fig. 15). These results indicated that, in contrast to numerous box C/D snoRNAs functioning as methylation guides, snR190 is required for optimal yeast proliferation.

In yeast, the few known examples of snoRNAs required for yeast proliferation (U3, U14, snR10, and snR30) function as chaperones in the 40S r-subunit biogenesis[10–13]. We next tested whether snR190 might function as a snoRNA chaperone in the maturation of pre-60S r-particles by analyzing pre-rRNA processing in the absence of snR190. Total RNA was extracted from snr190-[mut.C] strains and steady-state levels of pre-rRNA species were analyzed using northern blotting. Loss-of-function of snR190 resulted in a strong increase in the steady-state levels of the 35S and 33S/32S pre-rRNAs in the snr190-

[mut.C] mutant compared to the wild-type strains (Fig. 2a and Supplementary Fig. 5a for the W303 background; Supplementary Fig. 4a and Supplementary Fig. 5b for the BY4741 background), indicating that the maturation of early 90S r-particles was delayed. We further observed a slight increase in the 27SA$_2$ precursor levels and reduced accumulation of the 27SB pre-rRNAs. The 27SB/27SA$_2$ ratios therefore substantially decreased in the absence of snR190 in both genetic backgrounds (Supplementary Fig. 5c, d). We conclude that snR190 loss-of-function impairs conversion of 27SA$_2$ to 27SB pre-rRNA within pre-60S r-particles. Low-molecular-weight rRNAs were also studied to determine whether the absence of snR190 also influences production of the downstream intermediates. The levels of the 7S pre-rRNAs did not show any apparent change (Supplementary Fig. 6a, b) nor did the levels of the mature 5.8S rRNAs (Supplementary Fig. 7a, b). Furthermore, no change in the level of 20S pre-rRNA was observed in absence of snR190, suggesting that snR190 is not involved in pre-40S r-particle maturation. Importantly, loss-of-function of snR190 slightly affected production of the mature 25S rRNA (Fig. 2a and Supplementary Fig. 4a). These phenotypes are not due to off-target mutations introduced by the CRISPR-Cas9 approach, as ectopic re-expression of snR190 in the snr190-[mut.C] strain partially rescued the pre-rRNA processing defects as shown by the reduced levels of the 35S and 27SA$_2$ intermediates (Fig. 2b). Although these levels did not drop to those observed in the wild-type W303 strain, accumulation of the 27SB intermediate appeared fully restored to wild-type levels (Fig. 2b and Supplementary Fig. 8a). Similar observations were made in the BY4741 background (Supplementary Fig. 8b and Supplementary Fig. 8c), indicating that although these phenotypes are relatively weak, they are highly reproducible.

To determine whether the pre-rRNA processing defects observed in the absence of snR190 impact 60S r-subunit biogenesis, we analyzed polysome profiles from cell extracts of the snr190-[mut.C] strains in both genetic backgrounds. We observed that loss-of-function of snR190 resulted in a minor deficit in 60S r-subunits, as judged from the slight reduction in the levels of free 60S relative to 40S r-subunits, and the appearance of half-mer polysomes indicative of a shortage in 60S subunit and/or translation defects[34,35] (Fig. 2c, W303 background). Re-expression of snR190 restored a close to wild-type polysome profile, clearly reducing the presence of half-mer peaks (Fig. 2d). Similar results were obtained in a strain derived from BY4741 (Supplementary Fig. 4b). These results indicate that loss-of-function of snR190 induces minor but significant pre-rRNA processing defects in the maturation pathway of the 25S rRNA, thus affecting production of 60S r-subunits and possibly also translation.

**snR190 does not function as a ribose 2'-O-methylation guide.** snR190 contains an antisense element upstream of its box D' that is complementary to a sequence embedded within the root helix (helix H73) of domain V of the 25S rRNA (box A, Supplementary Fig. 2a). This antisense sequence is predicted to guide methylation of G2395 in the 25S rRNA[36,37]. Several large scale studies have been undertaken to exhaustively detect and quantify rRNA methylations in yeast[26–28], but none of them detected a ribose 2'-O-methylation at position G2395. To verify these data with a specific emphasis on G2395 methylation, we mapped methylations in this region of the 25S rRNA using primer extensions at low dNTP concentration[38]. Two different primers, proximal and distal, were used in these experiments, located 19 and 40 nucleosides upstream of G2395 respectively (Supplementary

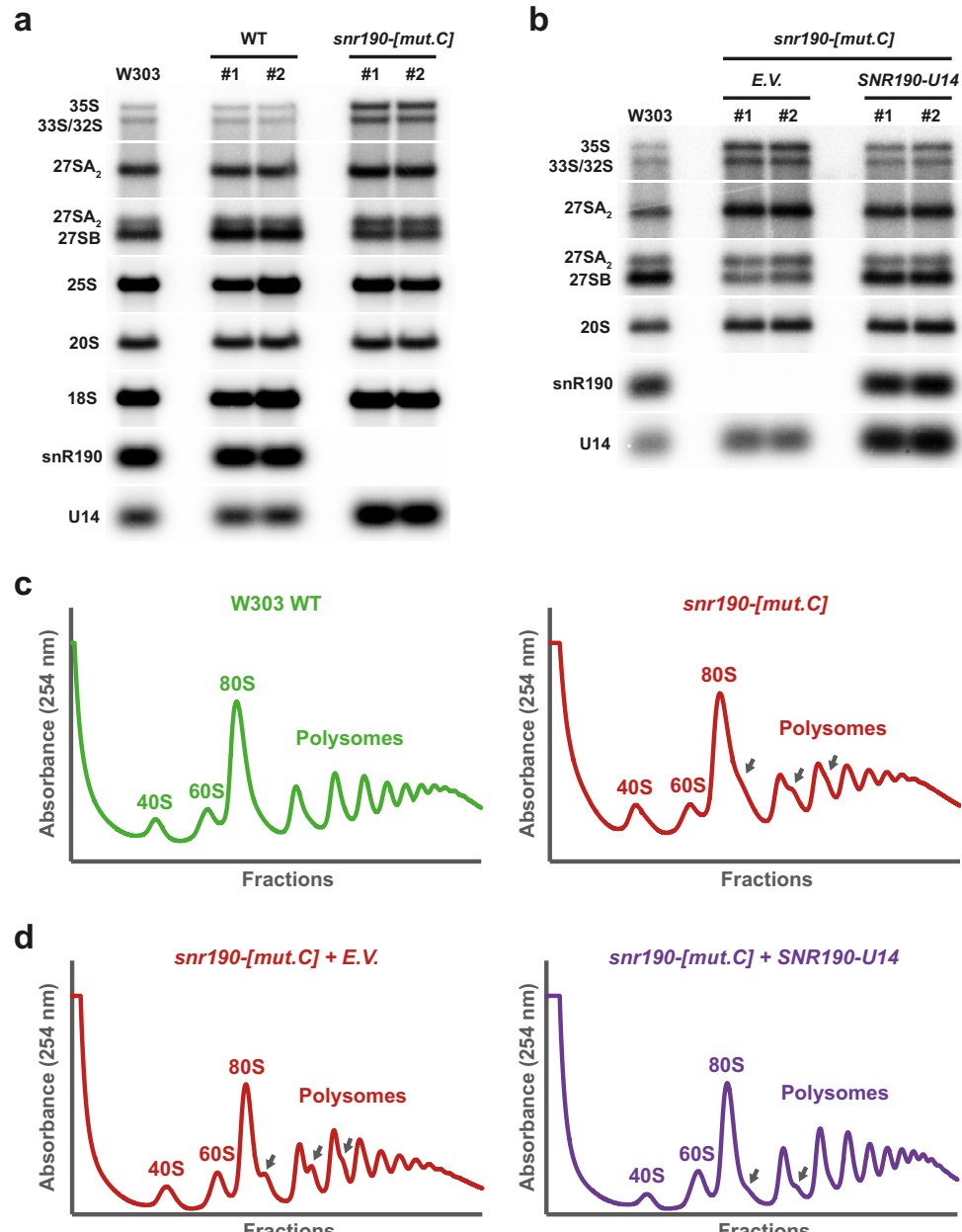

**Fig. 2 snR190 functions in pre-60S r-particle maturation. a** Accumulation levels of rRNA precursors in W303 wild-type (WT) and *snr190-[mut.C]* strains. Total RNA extracted from two independent clones (#1 and #2) for each strain was analyzed by northern blotting. The RNA molecules detected using radiolabeled oligonucleotide probes (Supplementary Table 4) are indicated on the left. **b** Accumulation levels of rRNA precursors in W303 *snr190-[mut.C]* strain and upon ectopic re-expression of snR190. Total RNA extracted from two independent clones (#1 and #2) of strain W303 *snr190-[mut.C]* transformed with a vector rescuing snR190 expression (*SNR190-U14*) or the corresponding empty vector as control (*E.V.*) was analyzed by northern blotting. The same probes as in **a** were used. **c** Polysome profiles on sucrose gradients of wild-type (green) and *snr190-[mut.C]* (red) strains in the W303 background. Total cellular extracts prepared from these strains were centrifuged through 10–50% sucrose gradients. A254 was measured during gradient fractionation. The identity of the different peaks is indicated and half-mers are marked with arrows. **d** Polysome profiles on sucrose gradients of W303 *snr190-[mut.C]* strain transformed with an empty vector (*E.V.*, red) or upon ectopic re-expression of snR190 (*SNR190-U14*, purple). Total cellular extract preparation and analysis were carried out as in **c**.

Fig. 9). For this experiment, we used the *snr190-[mut.C]* strain (BY4741 background) transformed with the *SNR190-U14* expression plasmid (*WT*) or the empty vector (*E.V.*) as control. Primer extension using the distal primer revealed two expected signals corresponding to methylations of nucleosides U2417 and U2421 (Fig. 3, right panel). No clear signal located further upstream at G2395 was observed. Primer extension with the proximal primer (Fig. 3, left panel) confirmed the absence of a strong, snR190-dependent signal at position G2395. Analysis of

snR190 putative secondary structure revealed that a potential internal stem can be formed between part of the antisense sequence and a region of the snoRNA including box C' and the downstream sequence (Supplementary Fig. 2a), possibly inactivating the guide element. To test whether this internal stem prevents methylation of G2395, we introduced mutations in the *SNR190-U14* expression plasmid to destabilize the stem (*snr190-[mut.S]*, Supplementary Fig. 2b). These mutations did not provoke any change in the signal obtained at position G2395 (Fig. 3).

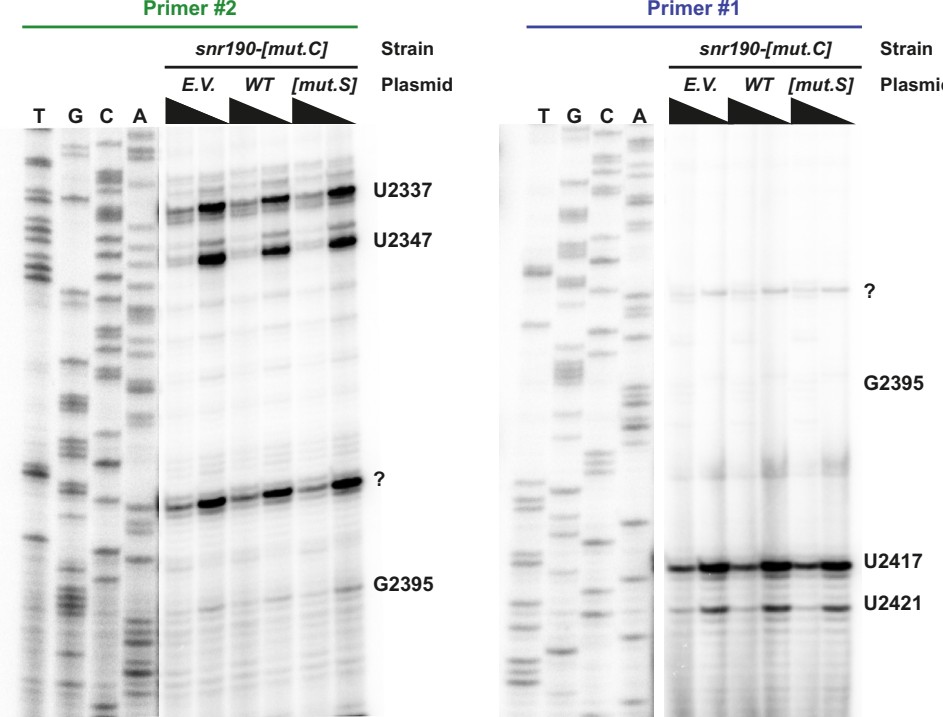

**Fig. 3 snR190 does not function as a methylation guide.** Reverse transcription experiment at normal or low dNTP concentration (black triangles) using total RNA extracted from strain BY4741 *snr190-[mut.C]* transformed with plasmids expressing the wild-type (*WT*) or *snr190-[mut.S]* versions of snR190 or the empty vector (*E.V.*) as control. Two different primers were used to initiate reverse transcription: primer #2 (proximal, green) and primer #1 (distal, blue). A sequencing ladder was loaded in parallel to identify the nucleosides inducing reverse transcription stops. Note: a weak reverse transcription stop was observed at position G2395 with the proximal primer, but this signal was detected both in the presence or absence of snR190, suggesting that it may correspond to a nonspecific signal rather than a bona fide methylation-induced reverse transcription stop. The question mark (?) indicates a strong reverse transcription stop that is not associated to currently known nucleoside methylation. These experiments were repeated two times independently with each primer and similar results were obtained.

We conclude that snR190 does not direct 25S rRNA modification at position G2395.

**snR190 functions as a chaperone in pre-60S r-particle maturation.** In addition to its antisense box A, snR190 contains another 15 nucleoside-long potential antisense element showing perfect complementarity to a sequence within domain I of the 25S rRNA, located in the vicinity of one of the Npa1 crosslinking sites[25] (box B, Supplementary Figs. 1 and 2a). In the mature 60S r-subunit, this 25S rRNA sequence is base paired to the 5.8S rRNA. Part of snR190 box B has been described previously to reinforce the base pairing of box A in the vicinity of G2395[39]. In any case, snR190 contains two long antisense elements with the potential to base-pair to two different regions of the 25S rRNA in domains I and V. In order to test whether boxes A and B are required for snR190 function in the maturation of pre-60S r-particles, we introduced drastic mutations in boxes A and/or B on the *SNR190-U14* expression vector described above (*snr190-[mut.A]*, *snr190-[mut.B]*, or *snr190-[mut.AB]*, Supplementary Fig. 2c). The resulting plasmids were used to transform the *snr190-[mut.C]* strain in the BY4741 or W303 genetic background. Mutation of box A or box B did not affect yeast cell proliferation as both mutant strains displayed a doubling time very similar to that of the *snr190-[mut.C]* strain rescued by the plasmid expressing wild-type snR190 (Supplementary Table 2 and Supplementary Fig. 15). In contrast, simultaneous mutation of boxes A and B induced a reproducible growth defect in both genetic backgrounds, with a doubling time intermediate between those of the wild-type and *snr190-[mut.C]* strains (Supplementary Table 2 and Supplementary Fig. 15). In no case was this the result

of reduced steady-state levels of the mutated snoRNAs (Fig. 4b). We next analyzed r-particle profiles in these mutants after fractionation of total cell extracts on density gradients. Mutation of boxes A or B alone showed polysome profiles very similar to that of the wild-type W303 strain (Fig. 4a). In contrast, mutation of both boxes A and B induced a minor imbalance of free 40S and 60S r-subunits, comparable to that observed in the *snr190-[mut.C]* strain, and the appearance of half-mer polysomes. Pre-rRNA processing analyses in the different strains gave results consistent with the polysome profiles (Fig. 4b, W303 background). The 27SB/27SA$_2$ ratio dropped to approximately half that of the wild-type in the absence of snR190 (Supplementary Fig. 8a), reflecting the accumulation of the 27SA$_2$ and depletion of the 27SB species. Ectopic expression of wild-type snR190, or the mutations of box A or box B alone, restored a close to wild-type 27SB/27SA$_2$ ratio. In contrast, mutation of both boxes resulted in a 27SB/27SA$_2$ ratio close to that of the *snr190-[mut.C]* strain (Fig. 4b and Supplementary Fig. 8a). These results were also reproduced in the BY4741 background (Supplementary Fig. 8b, c).

Altogether, these data show that individual mutation of box A or B does not negatively affect the function of snR190 whereas simultaneous mutation of both boxes induces pre-60S r-particle maturation defects.

**snR190 is retained in preribosomal particles in absence of Dbp7.** The genetic screen described in Fig. 1 suggests that Dbp7 may contribute to the dissociation of snR190 from its 25S rRNA target(s) within pre-60S r-particles. To further test this hypothesis, we used a strain expressing an HTP-tagged version of Nop7,

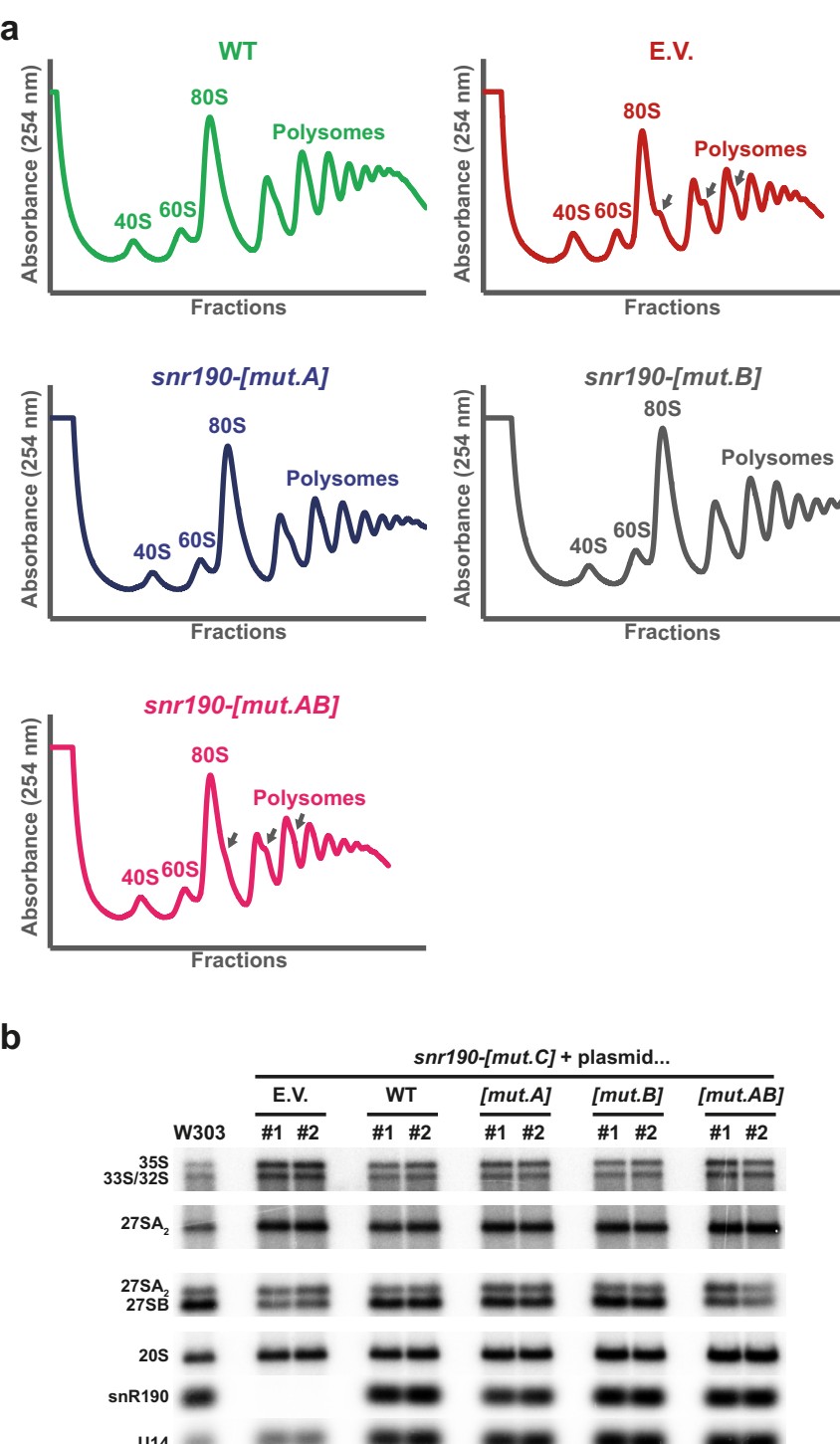

**Fig. 4 snR190 boxes A and B are required for its function in LSU synthesis. a** Polysome profiles on sucrose gradients of W303 *snr190-[mut.C]* strain transformed with vectors supporting expression of wild-type snR190 (WT, green) or bearing mutations in its antisense elements box A (*snr190-[mut.A]*, dark blue), box B (*snr190-[mut.B]*, grey) or both (*snr190-[mut.AB]*, pink), or with the empty vector as control (E.V., red). Cell extracts were prepared and analyzed as in Fig. 2c. **b** Steady-state levels of distinct pre-rRNAs and U14 and snR190 snoRNAs in isogenic wild-type and *snr190-[mut.C]* strains in the W303 background; cells were transformed with plasmids supporting expression of wild-type snR190 (WT) or bearing mutations in its antisense elements box A ([*mut.A*]), box B ([*mut.B*]) or both ([*mut.AB*]), or with the empty vector as control (E.V.). Experiments were performed as explained in the legend of Fig. 2b. Note: the first five lanes of this figure are identical to those presented in Fig. 2b, where they were used to support the conclusion that ectopic re-expression of snR190 restores pre-rRNA processing in the *snr190-[mut.C]* strain. This experiment was performed once with each independent clone for the W303 background and twice with each independent clone for the BY4741 background (see also Supplementary Fig. 8).

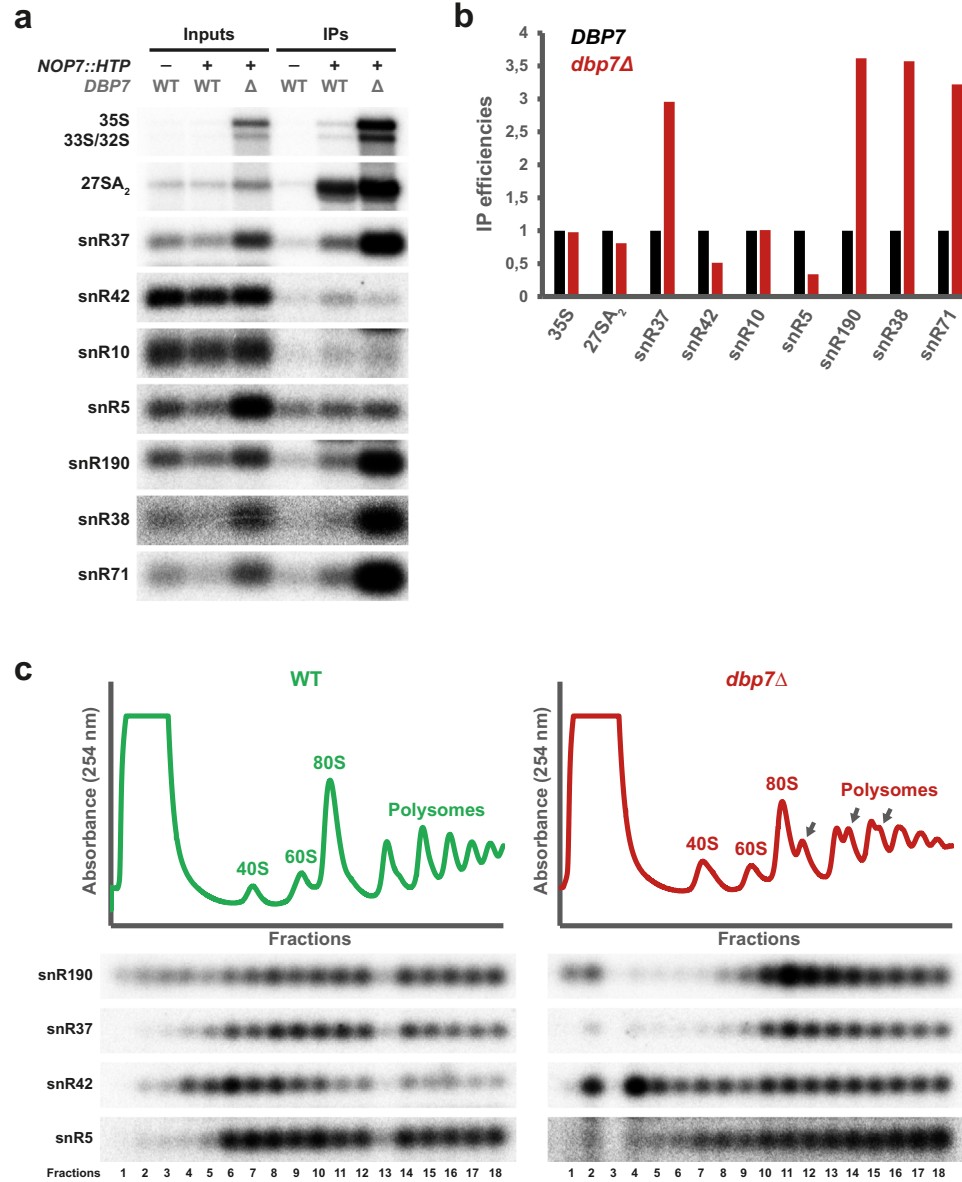

**Fig. 5 snR190 is retained into preribosomal particles in the absence of Dbp7. a** Immunoprecipitation of Nop7-HTP-containing preribosomal particles in the presence or absence of Dbp7. Particles were immunoprecipitated from total cell extracts prepared from wild-type or *dbp7Δ* strains expressing an HTP-tagged version of Nop7. As a background control, the wild-type strain expressing untagged Nop7 was similarly processed. Some rRNA precursors and snoRNAs (indicated on the left) present in the total extracts (Inputs) or in the immuno-precipitates (IPs) were analyzed by northern blotting using selected radiolabeled probes (Supplementary Table 4). **b** Quantification of the northern blot signals obtained for the hybridizations shown in **a** using PhosphorImager data and the MultiGauge software. The histogram represents the co-immunoprecipitation efficiencies (IPs over Inputs) of the indicated RNA species with Nop7-HTP in the presence (*DBP7*, black) or absence (*dbp7Δ*, red) of Dbp7. **c** Sedimentation profile of snR190, snR37, snR42, and snR5 snoRNAs in the presence or absence of Dbp7. Total cellular extracts prepared from the wild-type (WT, green) or *dbp7Δ* (*dbp7Δ*, red) strains were centrifuged through 10–50% sucrose gradients. A$_{254}$ was measured during gradient fractionation and the profiles are represented. RNAs extracted from the first 18 fractions were analyzed by northern blotting to detect the indicated snoRNAs. Note: the RNA samples corresponding to each gradient were processed separately (electrophoresis, transfer, hybridization, exposure, data acquisition). Therefore, the exposures presented for each snoRNA in the two strains have been chosen arbitrarily. Please refer to the intraseries quantifications (preribosome-bound versus free ratios) presented in Supplementary Fig. 10 for the interpretation of these data. This experiment has been performed once with the *dbp7Δ* strain and once with the strain expressing the Dbp7$_{K197A}$ catalytic mutant (Fig. 6c) with similar results for the snoRNAs tested.

a component of 90S and pre-60S r-particles[40,41] to purify these particles in both a *dbp7Δ* strain and its isogenic wild-type counterpart (Fig. 5a). Then, we compared by northern blotting the association of snR190 and other snoRNAs previously found efficiently crosslinked to Npa1 by CRAC[25] with these particles. As shown in Fig. 5a, b, snR190 clearly co-immunoprecipitated with Nop7-HTP in both tagged strains, but the efficiency of co-

immunoprecipitation was much higher in the *dbp7Δ* strain. In contrast, the box H/ACA snoRNAs snR5, snR42, and snR10 were weakly co-immunoprecipitated with Nop7-HTP in both conditions, although snR42 and snR10 base-pair to domain V of the 25S rRNA, close to one of the base-pairing sites of snR190. Strikingly, however, the box H/ACA snoRNA snR37, and the box C/D snoRNAs snR38 and snR71 were also preferentially retained

within the preribosomal particles containing Nop7-HTP in the *dbp7Δ* strain (Fig. 5a, b). These results indicate that the absence of Dbp7 alters the steady-state association of snR190 with early preribosomes, as well as that of some, but not all, snoRNAs interacting with domain V of the 25S rRNA.

To further confirm snR190 retention in 90S and pre-60S r-particles in the absence of Dbp7, the sedimentation profiles of snR190 on density gradients were compared in *dbp7Δ* and wild-type cells. Polysome profile analyses revealed that lack of Dbp7 caused a robust decrease in the amount of free 60S r-subunits (Fig. 5c) and formation of prominent half-mer polysomes, in agreement with the previously reported function of Dbp7 in 60S r-subunit production[29,30]. RNA was isolated from gradient fractions and the sedimentation profile of snR190 analyzed by northern blotting (Fig. 5c). In wild-type cells, snR190 sedimented both with free snoRNPs in the top fractions of the gradient and with preribosome-bound snoRNPs in denser fractions containing 90S and pre-60S r-particles. Deletion of *DBP7* induced a clear increase in the proportion of preribosome-bound snR190. Similarly, the sedimentation profile of snR37 also appeared affected to the same extent, whereas that of snR5 and snR42 did not show the same preribosome enrichment (Fig. 5c). Quantification of the signals detected in the top (1–5) or preribosome-bound (7–18) fractions of the gradients for each snoRNA confirmed an increase in the bound/free ratios for snR190 and snR37, but not for snR42 and snR5 (Supplementary Fig. 10a).

These data were further supported using sedimentations through low magnesium (low-$Mg^{2+}$) sucrose gradients, in which polysomes are dissociated into free 40S and 60S r-subunits. Compared to the wild-type control, lack of Dbp7 resulted in an increase in the proportion of the snR190, snR71, snR37, and snR38 snoRNAs co-sedimenting with pre-60S and 90S r-particles (Supplementary Fig. 11a). In contrast, the sedimentation profile of other snoRNAs such as snR42, snR5, snR10, snR61, and the 5 S rRNA, were less affected. Quantification of the radioactive signals detected in the different fractions of the gradient confirmed the increased sedimentation in fractions 9–13 of the gradient of snR190, snR71, snR37, and snR38 snoRNAs in absence of Dbp7 (Supplementary Fig. 11b).

As a member of the DExD-box family of putative ATP-dependent RNA helicases, Dbp7 is expected to possess ATPase activity, which may contribute to release of snR190 from the pre-rRNA. We tested in vitro the ATPase activity of recombinant wild-type Dbp7 protein (Dbp7$_{WT}$) and a mutant version carrying a lysine to alanine amino acid substitution in the conserved motif I (also referred to as Walker motif A) of the RecA1 domain (Dbp7$_{K197A}$). Recombinant His-ZZ tagged Dbp7$_{WT}$ and Dbp7$_{K197A}$ proteins were expressed and purified to apparent homogeneity from *Escherichia coli* (Fig. 6a) and their in vitro ATPase activity was tested using a NADH-coupled ATPase assay in the presence or absence of model RNA (Fig. 6b). We observed that Dbp7 on its own displayed a very weak ATPase activity, which was strongly stimulated by RNA. In contrast the Dbp7$_{K197A}$ mutant displayed a greatly reduced activity in the same conditions (Fig. 6b).

To determine whether the ATPase activity of Dbp7 is required in vivo for release of snR190 from preribosomes, a yeast strain expressing Dbp7$_{K197A}$ was generated. This strain showed a sedimentation profile of r-particles very similar to that of the *dbp7Δ* strain (Fig. 6c) but, compared to a wild-type strain, a milder growth defect was observed for the *dbp7Δ* strain (Supplementary Fig. 12). Analysis of the levels of free versus preribosome-bound snoRNAs showed a strong accumulation of snR190 into preribosomes in the strain expressing Dbp7$_{K197A}$, similar to that previously observed in the *dbp7Δ* strain. In contrast, the sedimentation profile of two other snoRNAs, snR10, and snR42, were not drastically changed upon expression of Dbp7$_{K197A}$. Quantification of the signals detected in the top or preribosome-bound fractions of the gradients clearly revealed an increase in the bound/free ratio for snR190 but not for snR10 and snR42 (Supplementary Fig. 10b). We conclude that the enzymatic activity of Dbp7 is required for efficient release of snR190 from preribosomal particles.

**snR190 mutations alleviate the growth defect of the *dbp7Δ* strain.** We hypothesized that the growth and processing defects observed in the *dbp7Δ* strain may be due to the aberrant retention of snR190 and other snoRNAs in early preribosomal particles. In order to test this, we eliminated snR190 in the *dbp7Δ* strain and analyzed the phenotypes of the resulting double mutant. The absence of snR190 slightly alleviated the growth defect of a *dbp7Δ* strain at 30 °C (Fig. 7a and Supplementary Table 2 and Supplementary Fig. 15), and the recovery was even more evident at 37 °C. Loss of snR190 also improved the growth rate of the strain expressing Dbp7$_{K197A}$ at 22, 25, and 30 °C (Supplementary Fig. 12). This result indicates that the absence of snR190 alleviates the phenotypes resulting from the loss of the ATPase activity of Dbp7, in line with the model that Dbp7 contributes to the release of snR190 from its base-pairing sites with the pre-rRNA.

We next determined whether weakening of the base pairing between snR190 and the 25S rRNA was responsible for this phenotypic rescue. We expressed in the *dbp7Δ snr190-[mut.C]* strain, different mutant versions of snR190 bearing either multiple mutations in box A (*snr190-[mut.A]*) or a single substitution of snR190 guanosine 34 into a cytidine (*snr190-[mut.G34C]*), to mimic the mutation on the 25S rRNA (C2392U) isolated in our suppressor screen (Fig. 1). In both cases, the growth defect of the *dbp7Δ* strain was partially alleviated (Supplementary Table 2 and Supplementary Fig. 15).

Analysis of pre-60S r-particle maturation in these strains showed rescue effects consistent with the growth data. As previously described[30], the *dbp7Δ* strain showed an increase in the steady-state levels of the 35S and the 27SA$_2$ precursors and a marked decrease of the 27SB pre-rRNAs (Fig. 7b, c). We observed a partial restoration of 27SB precursor levels in the double *dbp7Δ snr190-[mut.C]* mutant and upon mutagenesis of box A. In particular, the single G34C substitution in snR190 (*snr190-[mut.G34C]*) partially restored production of the 27SB intermediates when introduced in the *dbp7Δ* strain (Fig. 7b, c). These data indicated that the loss of snR190 directly alleviates the 27SB pre-rRNA production defects arising in the absence of Dbp7. Interestingly, however, 35S pre-rRNA accumulation was not restored to the wild-type level in these strains, which could reflect additional snR190-independent functions of Dbp7 (see Discussion).

Altogether, these data show that the lack of snR190, or mutations weakening the base pairing between its box A and pre-rRNA, partially suppress the growth and pre-rRNA processing defects of a *dbp7Δ* strain. Thus, the function of Dbp7 becomes less limiting in these conditions, strongly suggesting that the snR190-pre-rRNA duplex is a cellular substrate of Dbp7.

**Stable association of the Npa1 complex with preribosomes requires snR190.** snR190 is the snoRNA most efficiently cross-linked to Npa1[25]. To understand further this relationship, we next studied the impact of the lack of snR190 and Dbp7 on the association of the Npa1 complex with nascent preribosomes. To specifically purify early 90S and pre-60S r-particles containing the Npa1 complex, we fused Noc1, another known component of these early r-particles[42], to the Tandem-Affinity Purification

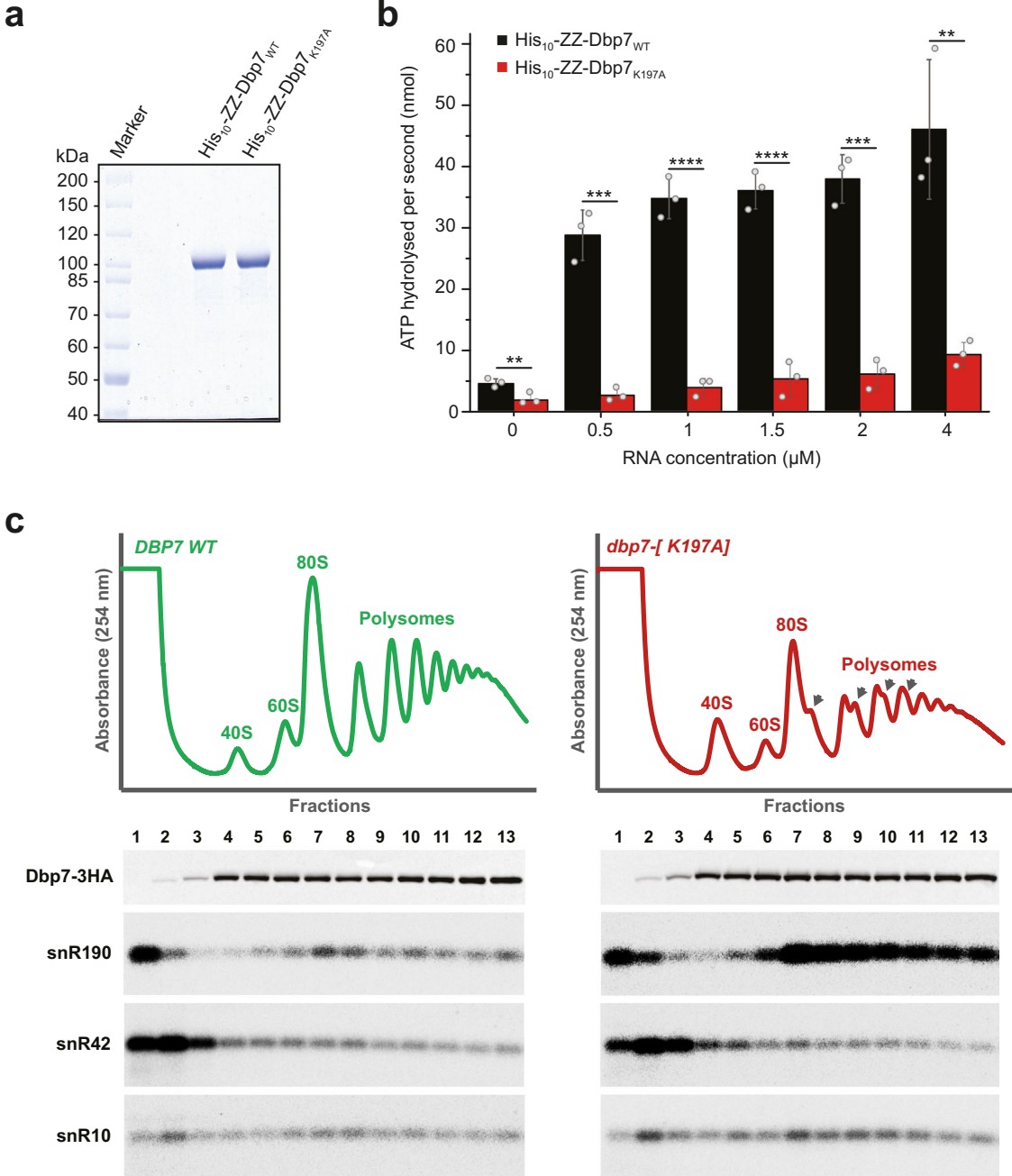

**Fig. 6 The ATPase activity of Dbp7 is required for snR190 release from preribosomal particles. a** His$_{10}$-ZZ-tagged Dbp7$_{WT}$ and Dbp7$_{K197A}$ were recombinantly expressed in *E. coli* and purified by nickel-affinity chromatography. Purified proteins were separated by denaturing SDS-PAGE and visualized by Coomassie staining. **b** The rate of ATP hydrolysis by purified His$_{10}$-ZZ-Dbp7$_{WT}$ (black) and Dbp7$_{K197A}$ (red) was monitored using an in vitro NADH-coupled ATPase assay in the presence of increasing amounts of RNA. Data from three independent experiments are presented as mean ± standard deviation. Statistically significant differences determined using one-tailed Student's *t*-test are indicated by asterisks (0 μM RNA: **$p < 0.01$ (0,008696); 0.5 μM RNA: ***$p < 0.001$ (0,000217); 1 μM RNA: ****$p < 0.0001$ (0,000061); 1.5 μM RNA: ****$p < 0.0001$ (0,000106); 2 μM RNA: ***$p < 0.001$ (0,000139); 4 μM RNA: **$p < 0.01$ (0,002657). **c** Sedimentation profile of Dbp7-3HA and snR190, snR42, and snR10 snoRNAs in cells expressing Dbp7$_{K197A}$. Total cellular extracts prepared from the *dbp7Δ* strain transformed with vectors encoding 3HA-tagged versions of Dbp7$_{WT}$ (green) or Dbp7$_{K197A}$ (red) were centrifuged through 10–50% sucrose gradients. A$_{254}$ was measured during gradient fractionation and the profiles are represented. Proteins and RNAs extracted from the first 13 fractions were analyzed by western blotting and northern blotting, respectively, to detect wild-type or mutant Dbp7-3HA proteins and snoRNAs. This experiment has been performed once with the strain expressing the Dbp7$_{K197A}$ catalytic mutant and once with the *dbp7Δ* strain (Fig. 5c) with similar results for the snoRNAs tested.

(TAP) tag in wild-type (BY4741), *snr190-[mut.C]*, and *dbp7Δ* strains. The RNAs and proteins co-purified with Noc1-TAP in each condition were analyzed by northern and western blotting, respectively (Fig. 8). In the absence of snR190, all tested members of the Npa1 complex, namely Npa1, Dbp6, Nop8, and Rsa3, were

less efficiently co-purified with Noc1-TAP compared to the wild-type conditions. This effect was not due to a lesser immunoprecipitation efficiency of early preribosomes as the levels of the 35S, 33S/32S, and 27SA$_2$ precursors were slightly higher in the Noc1-TAP-purified sample obtained from the *snr190-[mut.C]* strain, as

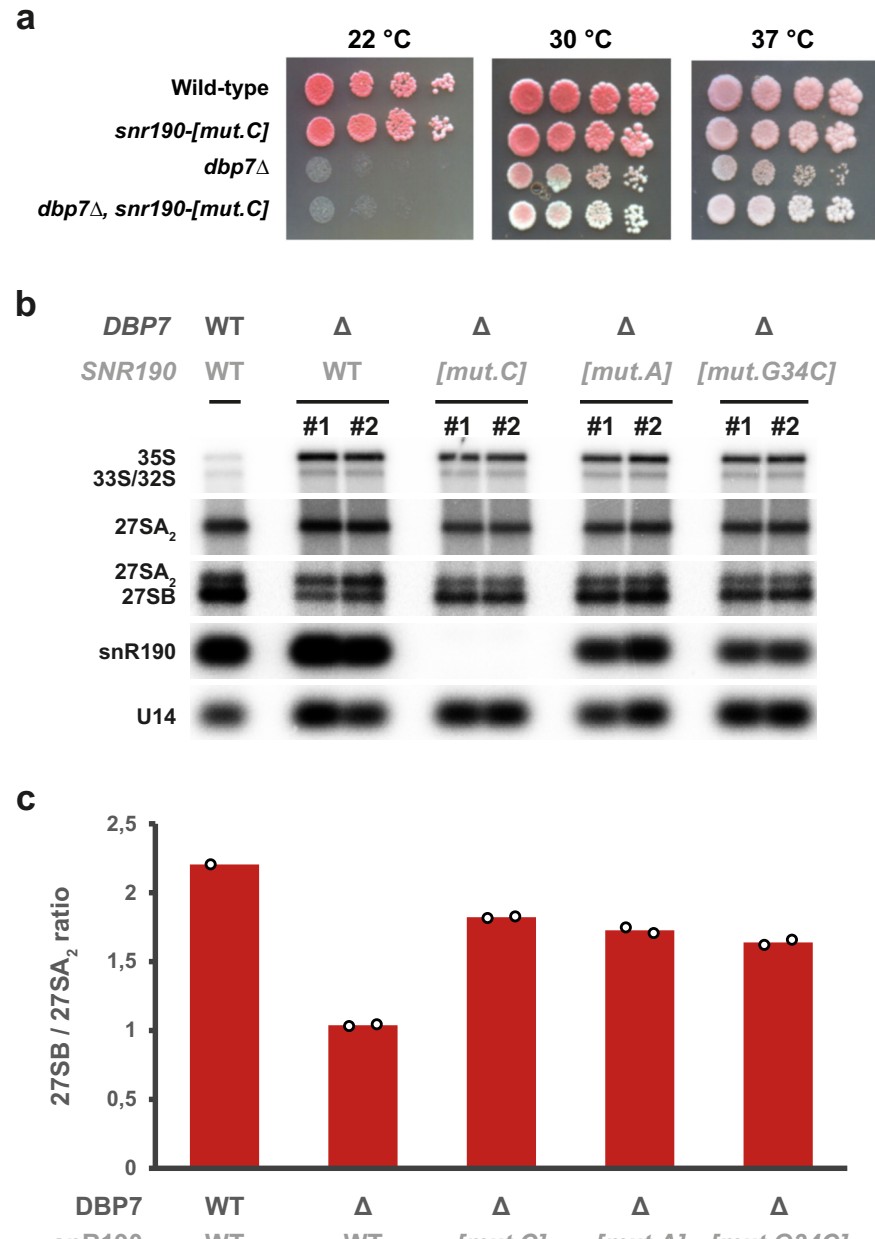

**Fig. 7 Mutations in *SNR190* alleviate the phenotypes of the *dbp7Δ* strain. a** Growth assay of isogenic wild-type, single *snr190-[mut.C]*, single *dbp7Δ* and double *dbp7Δ*, *snr190-[mut.C]* mutants in the W303 background. Serial dilutions of these strains were grown on YPD plates at the indicated temperatures for 3–5 days. **b** Accumulation levels of rRNA precursors in the wild-type BY4741 strain or in two independent clones (#1 and #2) of the *dbp7Δ* strain and the *dbp7Δ* strain further manipulated using the CRISPR-Cas9 approach to deplete snR190 (*snr190-[mut.C]*) or to express *snr190-[mut.A]* and *snr190-[mut.G34C]*. Experiments were performed as explained in the legend of Fig. 2b. **c** Quantification of the 27SB/27SA$_2$ ratios for the indicated strains from the precursor levels measured in **b** using PhosphorImager data and the MultiGauge software. Data correspond to two biological replicates. Histograms represent the mean values. The individual data points are shown.

were the levels of the box H/ACA snoRNP protein Nhp2 and the RNA helicase Prp43. We conclude that snR190 is required for efficient recruitment of the Npa1 complex into nascent preribosomes or for its stable association. In contrast, we observed opposite effects in the absence of Dbp7, namely that members of the Npa1 complex were more efficiently co-purified with Noc1-TAP compared to the wild-type control. Altogether, these results suggest that snR190 facilitates recruitment of the Npa1 complex into early preribosomal particles. Once the functions of snR190 and the Npa1 complex have been fulfilled, the RNA helicase Dbp7 promotes removal of the snR190 snoRNP from its target

site(s) in pre-60S r-particles, which in turn induces or facilitates the dissociation of the Npa1 complex and other snoRNPs.

## Discussion

Eukaryotic ribosome synthesis relies on the function of numerous DExD/H-box ATPases. In yeast, most of these enzymes are essential for cell viability, indicating that each fulfills a specific, nonredundant function in r-subunit maturation. Yeast genetics proved highly valuable in revealing functionally relevant links between several DExD/H-box ATPases and their substrates[43–46].

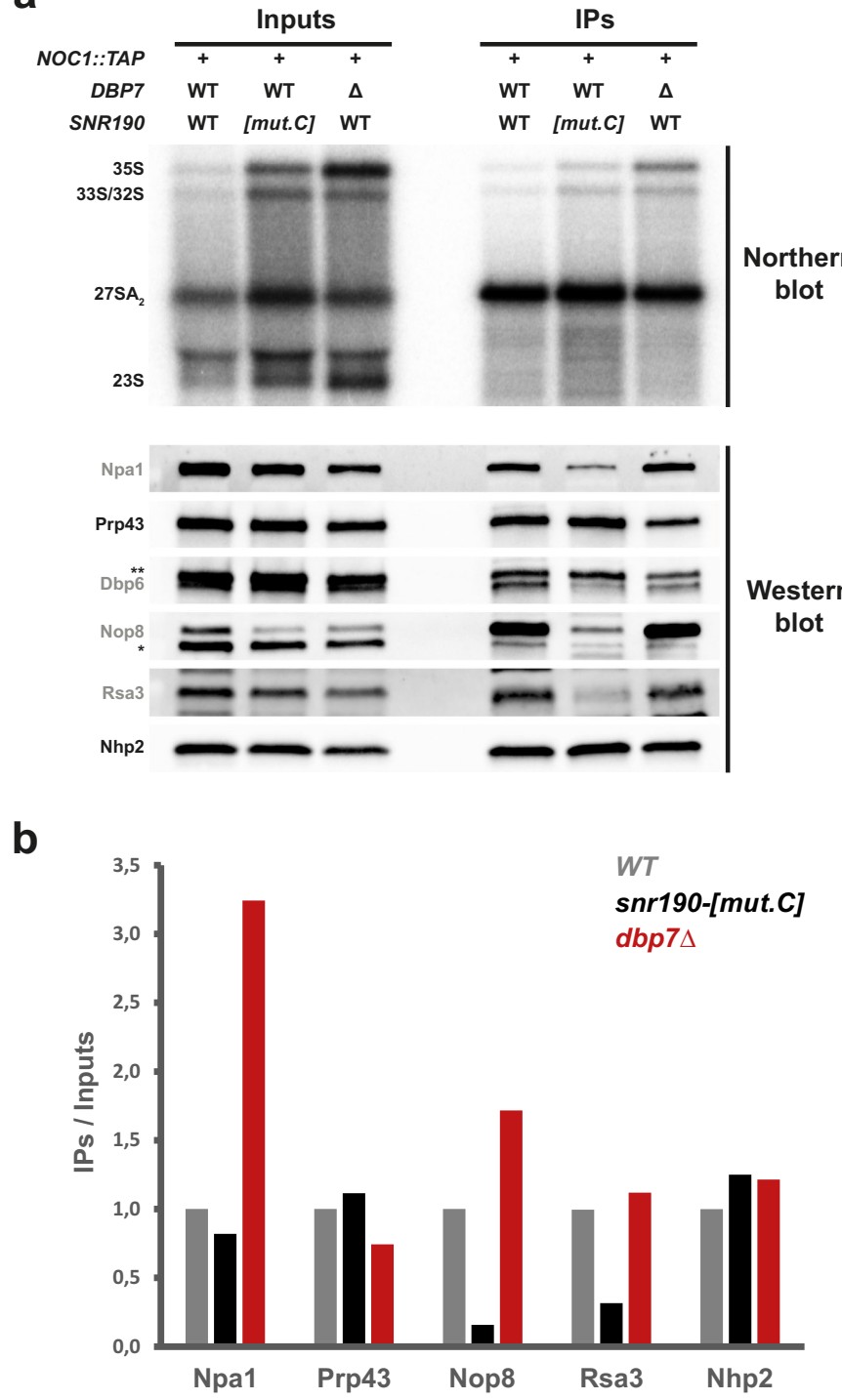

**Fig. 8 snR190 is required for stable incorporation of the Npa1 complex into preribosomes. a** Immunoprecipitation of Noc1-TAP-containing preribosomal particles from wild-type, *snr190-[mut.C]* and *dbp7Δ* mutant strains, and analysis of their RNA and protein contents by northern and western blotting, respectively. Particles were immunoprecipitated from total cell extracts prepared from wild-type, *snr190-[mut.C]*, and *dbp7Δ* strains expressing a TAP-tagged version of Noc1. Pre-rRNA components of 90S and early pre-60S r-particles (indicated on the left) present in the total extracts (Inputs) or in the immuno-precipitates (IPs) were analyzed by northern blotting (upper panel) using a radiolabeled probe (Supplementary Table 4). Protein levels of components of the Npa1 complex (indicated in grey) or other proteins present in early preribosomal particles (labeled in black) were analyzed by western blotting on the Input or IP samples (lower panel) using specific antibodies. *: nonspecific signal detected with the anti-Nop8 antibodies. **: Prp43 signal resulting from incubation of the membrane with anti-Prp43 antibodies (see above panel) prior to incubation with anti-Dbp6 antibodies. **b** Quantification of the western blot signals obtained in **a**. Western blot signals were quantified from ChemiDoc images (Biorad) using the Image Lab software (Biorad). The histogram represents the co-immunoprecipitation efficiencies (IPs over Inputs) of the indicated proteins with Noc1-TAP in the wild-type (*WT*, gray), *snr190-[mut.C]* (black), and *dbp7Δ* (red) strains, normalized with respect to the ratios in the wild-type strain. Note: the Dbp6 signal obtained in **a** could not be quantified accurately due to the proximity of the Prp43 signal.

The data presented herein strongly suggest that Dbp7 is the enzyme responsible for the dissociation of snR190 from the pre-rRNA to facilitate maturation of early pre-60S r-particles. Our genetic screen shows that a single C-to-U transition in the 25S rRNA sequence predicted to base-pair with snR190 partially alleviates the growth defect of the *dbp7Δ* strain. This mutation may impact ribosome synthesis and function in two ways. On the one hand, it creates an additional U-A base-pair within helix H73 of the 25S rRNA (Supplementary Fig. 1) that therefore likely stabilizes this helix, with potential consequences on pre-60S r-particle maturation and/or translation. On the other hand, it weakens the base pairing between snR190 box A and the 25S rRNA sequence by replacing a canonical C–G base-pair by a wobble U•G base-pair. It is difficult to determine which of these two aspects, or whether both simultaneously, are responsible for the suppression effect. Although the EMY65 strain used in our genetic screen displays a severe growth defect compared to conventional strains as ribosome synthesis is impaired, we still analyzed the sedimentation profile of snR190 in the presence or absence of the 25S rRNA C2392U suppressor mutation (Supplementary Fig. 13a, b). We observed a slight but reproducible decrease in the retention of snR190 into preribosomes in this strain grown on a glucose-containing medium (Dbp7 depletion) in the presence of the C2392U mutation. This phenotype is in line with the partial suppression of the pre-rRNA processing (Supplementary Fig. 14a, b) and growth (Fig. 1a) defects of the strain in these conditions. This result suggests that the C2392U mutation weakens the snR190/pre-rRNA base pairing. This hypothesis is reinforced by the fact that lack of snR190 or mutations that weaken its base pairing with the pre-rRNA also suppress the growth and pre-rRNA processing defects of the *dbp7Δ* strain in a conventional genetic background.

Yeast 25S rRNA G2395 ribose 2'-*O*-methylation has not been detected by several different high-throughput approaches[26–28], what we confirmed using directed analyses (Fig. 3). The molecular rationale behind this remains unclear as snR190 contains a C'/D' motif and antisense element (box A). The internal stem mobilizing part of box A within snR190 does not seem to be the reason (Fig. 3). snR190 is efficiently immunoprecipitated with Nop1, the methyltransferase of C/D-type snoRNPs[47]. Still, as snR190 contains both a C/D and a C'/D' motif and its box A is linked to the C'/D' motif, it could be that only the C/D motif bound by the core proteins ensures proper accumulation of the snoRNP and that the core proteins are not positioned properly on the C'/D' motif to catalyze the G2395 methylation. In line with this, sequence alignment of snR190 sequences from different yeast species revealed that boxes C' and D' of snR190 diverge from the consensus sequences[39]. Furthermore, in some species, box C' can be interrupted by several intervening nucleosides and may not be active[39]. Thus, we favor the hypothesis that the C'/D' motif of snR190 does not allow proper positioning of core proteins and thus precludes catalysis of the methylation.

Our data support the hypothesis that snR190 functions as a snoRNA chaperone during pre-60S r-subunit maturation. Loss-of-function of snR190 affects the conversion of the 27SA$_2$ pre-rRNA into 27SB. This is correlated with reduced large r-subunit production and the formation of half-mer polysomes. snR190 contains two antisense elements, boxes A and B, with the potential to base-pair with domains VI and I of the 25S rRNA, respectively. Two hypotheses can be formulated concerning the mode of action of snR190 in chaperoning the pre-rRNA during pre-60S r-particle maturation. First, in cooperation with the Npa1 complex, snR190 may establish a physical bridge between the 5' and 3' regions of the 25S rRNA to promote or stabilize their clustering during compaction of pre-60S r-particles. This clustering is a crucial early event in pre-60S r-particle maturation[20]

that is evolutionarily conserved as in bacteria, the 5' and 3' regions of large subunit rRNA are brought in close proximity through formation of a stem between flanking sequences on the RNA precursor[48]. This clustering may implicate a complex interplay between assembly factors, such as the Npa1 complex and snoRNAs in eukaryotes. In agreement with this hypothesis, our results show that the absence of snR190 affects the incorporation or stable association of the Npa1 complex within early preribosomal particles. Second, snR190 shields 25S rRNA sequences complementary to its antisense boxes A and B to prevent them from establishing nonfunctional base-pairing interactions. In line with this, chemical probing experiments suggested that within the 35S pre-rRNA, the 5' end of the 5.8S rRNA is base paired with ITS1 sequences[41,49], although this has been later challenged[21]. Processing events in the ITS1 are expected to abrogate this base pairing and release the 5' region of the 5.8S rRNA for base pairing with the 25S rRNA. As these conformational changes are expected to occur relatively late during the maturation pathway, after processing at site B$_{1S}$ and B$_{1L}$, we propose that snR190 may function as a placeholder to shield the 25S rRNA region until the 5' end of the 5.8S rRNA becomes available for base pairing. Our data show that inactivation of either box A or box B does not induce significant growth or processing defects, whereas mutation of both boxes simultaneously is more deleterious. This observation better supports a role of snR190 as a placeholder shielding its complementary sequences in the 5' and 3' regions of the 25S rRNA. A defect in shielding of any of the two sequences seems benign in terms of growth and processing, but a cumulative defect may become more deleterious.

Our genetic screen showed that a C-to-U transition in the 25S rRNA predicted to weaken the base pairing with snR190 alleviates the growth defect resulting from Dbp7 depletion. We showed in addition that the absence of snR190 or mutations within its antisense box A also alleviates the growth and processing defects of the *dbp7Δ* strain. Importantly, a single-nucleoside substitution within snR190 box A, predicted to be synonymous to the mutation identified in the 25S rRNA in the genetic screen, also partially restores the growth of the *dbp7Δ* strain. We therefore conclude that Dbp7 becomes more dispensable in the presence of mutations weakening the base pairing between snR190 and the pre-rRNA, suggesting that Dbp7 may utilize its catalytic activity to resolve the snR190 box A/pre-rRNA duplex. In line with this model, a CRAC analysis of Dbp7 binding sites described in a parallel study[50] revealed that Dbp7 binds to both snR190 and domain V of the 25S rRNA, in the vicinity of the base-pairing site of snR190 box A. Biochemical evidence also supports the model that Dbp7 dissociates snR190 from pre-60S r-particles. snR190, but not several other snoRNAs predicted to interact with the pre-rRNA in the PTC area, such as snR10 or snR42, is retained within preribosomal particles in the absence of Dbp7 (Fig. 5). However, several canonical modification guide snoRNAs (e.g., snR37, snR38, and snR71) targeting nucleosides of the PTC region are also retained within pre-60S r-particles in the absence of Dbp7, as are also most members of the Npa1 complex. As our genetic screen revealed a clear link between snR190 and Dbp7, the accumulation of the other snoRNAs is likely an indirect consequence of a defect in snR190 release.

The phenotypes resulting from lack of snR190 are less severe than those observed in the absence of Dbp7 (for a direct comparison of the defects, see Supplementary Fig. 6a–c). As the absence of Dbp7 induces the retention of several snoRNAs other than snR190 within 90S or pre-60S r-particles (Fig. 5), this cumulative retention could simply be the reason for the more severe defects observed upon deletion of *DBP7* than upon inactivation of snR190. It is possible that, without the function of

Dbp7, resolving pre-60S r-particles with several trapped snoRNAs might be more challenging for cells than generating spontaneously helix H73 in the absence of snR190. In any case, whether or not Dbp7 is actively or indirectly involved in the displacement of those abovementioned snoRNAs from the pre-rRNA remains to be solved. Another hypothesis to explain the more severe phenotypes of the *dbp7Δ* strain could be that Dbp7 fulfills other functions in the maturation of 90S and pre-60S particles, such as modulating the stability of intramolecular duplexes of the 25S rRNA or displacing protein–RNA interactions. Dbp7 CRAC data indicated that Dbp7 interacts with domain VI of the 25S rRNA in addition to domain V, suggesting that it may also intervene in remodeling events in this domain of the 25S rRNA within 90S or pre-60S r-particles[50].

## Methods

**Yeast strains and media**. All yeast strains used in this study were derivatives of either *S. cerevisiae* strains BY4741 (*MATa, his3Δ1, leu2Δ0, met15Δ0,* and *ura3Δ0*) or W303-1A (*MATa, leu2-3, 112 trp1-1, can1-100, ura3-1, ade2-1,* and *his3-11,15*). They are listed in the Supplementary Table 3. Strains were grown either in YP medium (1% yeast extract, 1% peptone) (Becton-Dickinson) supplemented with 2% glucose as the carbon source (YPD, rich medium) or in synthetic medium (0.17% yeast nitrogen base (MP Biomedicals), 0.5% $(NH_4)_2SO_4$ supplemented with 2% glucose (SD, minimal medium) or 2% galactose (SGal, minimal medium) and the required amino acids. Selection of the kanamycin-resistant transformants was done by addition of G418 to a final concentration of 0.2 mg/ml.

The strain expressing Nop7-HTP (His-tag-Tev cleavage site-ZZ sequence derived from *Staphylococcus aureus* Protein A) was obtained by transforming BY4741 with a PCR cassette produced using plasmid pBS1539-HTP and primers listed in Supplementary Table 4. Clones having integrated the *URA3* gene were selected on SD-Ura medium.

**Genome editing using CRISPR-Cas9**. In yeast, snR190 is expressed from a dicistronic precursor transcript (UMO1), also supporting expression of U14, an essential box C/D snoRNA that functions as an RNA chaperone in the maturation of the 18S rRNA[32]. Both snoRNAs are co-transcribed by RNA Pol II from the same promoter and then converted to mature snoRNAs by a series of endo and exonucleolytic processing events[32]. This particular genomic context precluded using conventional genomic knockout approaches to inactivate snR190 expression without affecting that of U14. To interfere with snR190 expression post-transcriptionally, we introduced point mutations in the genomic sequences encoding box C and the terminal stem of the snoRNA (*snr190-[mut.C]*, Supplementary Fig. 2b) using the CRISPR-Cas9 approach[33]. As previously described, these mutations are predicted to prevent assembly of C/D-type snoRNP core proteins and result in the exonucleolytic degradation of snR190 during maturation of the dicistronic precursor, without interfering with U14 production[51].

To generate the *snr190-[mut.C]* strain, the Cas9 expressing vector pML104 was first digested with *Swa*I (New England Biolabs) overnight at 25 °C. The enzyme was then inactivated at 65 °C for 20 min. *Bcl*I (Fermentas) was subsequently added to the reaction for 2 h at 50 °C. In order to target *SNR190* box C, oligonucleotides OHA494 and OHA495 (Supplementary Table 4) were designed using the online CRISPR Toolset (http://wyrickbioinfo2.smb.wsu.edu/crispr.html) and were hybridized at a concentration of 3 μM in T4 DNA Ligase Buffer. The hybridization was performed using a thermocycler: the oligonucleotide mix was heated at 95 °C for 6 min and the temperature was then gradually decreased to 25 °C at a rate of 1 °C per min. The hybridized oligonucleotides were then ligated to the digested pML104 plasmid overnight at 16 °C. The ligated vector was then transformed into competent *E. coli* cells (Stellar, Clontech). *SNR190* guide sequence insertion was confirmed by sequencing the resulting plasmids using the OHA497Bis primer (Supplementary Table 4). Yeast cells were then transformed by a one-step-transformation procedure with the single guide-containing plasmid along with a single-stranded donor oligonucleotide containing the desired *SNR190* box C mutations as well as mutations in the PAM sequence (OHA496, Supplementary Table 4). For a better transformation efficiency, carrier DNA (fish sperm DNA) was added and the transformation reaction was incubated for 45 min at 30 °C. Transformants were then selected on SD-Ura medium and the Cas9 plasmid was then eliminated by plating clones on medium containing 5-fluoroorotic acid (5-FOA) before being streaked on YPD plates. Genomic DNA from the resulting colonies was purified, the *SNR190* locus was amplified by PCR using (OHA497 and OHA498 primers, Supplementary Table 4) and the resulting PCR fragments sequenced with the same primers to verify the mutations. The same protocol was applied to generate, in the *dbp7Δ* strain, loss-of-function mutations in *SNR190* (*snr190-[mut.C]*), and introduce in its box A multiple mutations (*snr190-[mut.A]*) or the G34C mutation (*snr190-[mut.G34C]*) (the corresponding oligonucleotides are listed in Supplementary Table 4). Most of the conclusions of this study stem from the analysis of two independent mutant clones in two different genetic

backgrounds (BY4741 and W303), and two biological replicates of the experiments have been performed in each case.

**Plasmid constructions**. Ectopic expression of snR190 in the *snr190-[mut.C]* strain was achieved using the pCH32 plasmid[52] containing the genomic *SNR190-U14* expression cassette. This cassette was amplified by PCR using primers OMJ122 and OMJ123 (Supplementary Table 4) and inserted into the pCH32 plasmid previously digested with *Eco*RI and *Eco*RV, using the In-Fusion system (Clontech). The In-Fusion reaction was transformed into competent *E. coli* cells (Stellar, Clontech) and recombinant clones were sequenced using the OHA498 primer (Supplementary Table 4).

Introduction of mutations in box A (*snr190-[mut.A]*), box B (*snr190-[mut.B]*) or the internal stem (*snr190-[mut.S]*) of *SNR190* in the pCH32 plasmid containing the *SNR190-U14* expression cassette was achieved using the PCR-based In-Fusion mutagenesis system and appropriate primers for each mutation (Supplementary Table 4). The original template plasmid was then eliminated by the "Cloning Enhancer" treatment (Clontech) and the linear PCR products were circularized using the In-Fusion system before transformation into competent *E. coli* cells (Stellar, Clontech). All resulting plasmids were verified by sequencing. To generate a *SNR190-U14* expression vector bearing mutations in boxes A and B of snR190 (*snr190-[mut.AB]*), mutations in box B were introduced using the same protocol on the plasmid already harboring the *snr190-[mut.A]* mutation.

Introduction of the C2392T substitution in plasmid pNOY373 or reversion of T2392 to the wild-type cytosine in plasmids Sup. #2 or Sup. #10 was achieved as described above using the PCR-based In-Fusion mutagenesis system and appropriate primers for each mutation (Supplementary Table 4).

**Genetic screen for rRNA suppressors of Dbp7 loss-of-function**. To obtain rRNA suppressors able to bypass the growth defect of Dbp7 depletion, we first integrated at the *DBP7* locus, a *GAL::HA-DBP7* construct in the strain NOY891 (*MATa ade2-1 ura3-1 leu2-3 leu2-112 his3-11 can1-100 rdnΔΔ::HIS3*). In this strain (a generous gift from Prof. Nomura's lab), the chromosomal rDNA repeats are completely deleted (*rdnΔΔ*) and the sole source of mature rRNAs is a single rDNA repeat expressed from the 2 μ plasmid pNOY353, in which expression of 35S pre-rRNA is under the control of a galactose-inducible $GAL7$ promoter ($P_{GAL7}$-35S rDNA, 5S rDNA, *TRP1*)[31]. The chromosomal *DBP7* gene was replaced by a *GAL1::HA-DBP7* construct by transformation, which was generated by the one-step PCR strategy on the pFA6a-KanMX6-$P_{GAL1}$-3HA plasmid[53]. This strain was named EMY65. The presence of the *GAL1::HA-DBP7* allele was confirmed by PCR and DNA sequencing. Western blotting also confirmed the presence of a single protein band of the expected molecular mass when candidates are grown in SGal medium (Supplementary Fig. 3).

Then, we performed random mutagenesis of pNOY373, a 2 μ *LEU2* plasmid carrying a single rDNA repeat expressing both the 35S and 5S pre-rRNAs from their cognate promoters. For mutagenesis, the *E. coli* mutator strain XL1-Red (Stratagene) was used, following the procedure described by the manufacturer. About 15 independent mutageneses were performed, then, cells were pooled and plasmids purified by a standard midi-prep procedure. As the result of the mutagenesis, it was estimated that ca. 40% of the plasmid clones harbored loss-of-function mutations in the *LEU2* marker. The mutated plasmids were introduced into strain EMY65 and fast-growing transformants were screened on SD-His-Leu medium, where the expression of *GAL1*- and *GAL7*- driven promoters was repressed. The mutated pNOY373 plasmids were rescued from fast-growing candidates and used to retransform EMY65 strain in order to link the growth suppression to the presence of the mutagenized plasmid.

**Construction of a double *dbp7Δ snr190-[mut.C]* mutant**. To determine if the severe growth defect linked to the *dbp7* null allele could be partially suppressed by the loss-of-function of snR190, we performed a genetic analysis. To this end, the MCD1-7C strain (a *dbp7::HIS3MX6* strain complemented with a YCplac33-DBP7 plasmid)[30] was crossed to *snr190-[mut.C]* strain generated in the W303 background. The resulting diploid was sporulated, and tetrads were dissected. After dissection, as expected, the HIS3MX6 marker that deleted-disrupted the *DBP7* gene segregated $2^+:2^-$ in complete tetrads. Then, we counter-selected the *URA3*-marked YCplac33-DBP7 plasmid from several complete tetrads in plates containing 5-FOA. To identify tetratypes, we obtained total DNA from these tetrads and sequenced the genomic region containing the coding sequence of the *SNR190* gene. Two tetratypes were selected for further analyses.

**Detection of ribose-methylated nucleosides**. Ribose methylation was tested by reverse transcription at low dNTP concentration. Briefly, 5 μg of total yeast RNAs were mixed with 0.2 pmol of a radiolabeled oligonucleotide hybridizing downstream from the site of ribose methylation (primer #1 or #2). After heat denaturation (95 °C, 1 min), primer extension was carried out in the presence of either 1 mM (high concentration) or 0.1 mM (low concentration) dNTPs and with 10 units of AMV reverse transcriptase (Promega) according to the manufacturer's protocol. Sequencing of a plasmid containing the yeast 25S rRNA sequence (2333-2457) was performed with the USB® Sequenase™ version 2.0 DNA Polymerase Kit and used as a ladder.

**Sedimentation on sucrose gradients**. Yeast cells growing exponentially (500 ml cultures at $OD_{600}$ ~0.6) were treated for 10 min with 50 μg/ml cycloheximide (Sigma) added directly to the culture medium. Cells were collected by centrifugation, rinsed with buffer K [20 mM Tris-HCl pH 7.4, 50 mM KCl, 10 mM $MgCl_2$] supplemented with 50 μg/ml cycloheximide and collected again by centrifugation. Dry pellets were resuspended with approximately one volume of ice-cold buffer K supplemented with 1 mM dithiothreitol (DTT), 1× Complete EDTA-free protease inhibitor cocktail (Roche), 0.1 U/μl RNasin (Promega), and 50 μg/ml cycloheximide. About 250 μl of ice-cold glass beads (Sigma) were added to 500 μl aliquots of the resuspended cells which were broken by vigorous shaking, three times 2 min, separated by 2 min incubations on ice. Extracts were clarified through two successive centrifugations at 16,000 × g and 4 °C for 5 min and quantified by measuring absorbance at 260 nm ($A_{260}$). Equal amounts of extracts were loaded on 10–50% sucrose gradients in TMK buffer. Gradients were centrifuged at 260,800 × g for 2.5 h at 4 °C in a SW41 Ti rotor (Optima L-100 XP ultracentrifuge, Beckman Coulter). The gradient fractions were collected with a Foxy R1 gradient collector (Teledyne Isco) driven by PeakTrak software (Version 1.10, Isco Inc.). The $A_{254}$ was measured during collection with a UA-6 UV/VIS DETECTOR (Teledyne Isco).

For the fractionation experiment shown in Supplementary Fig. 11, we prepared r-subunits exactly as described previously[54]. Briefly, cell extracts were prepared in a buffer containing 50 mM Tris-HCl, pH 7.4, 50 mM NaCl, and 1 mM DTT in the absence of cycloheximide. Ten $A_{260}$ units were loaded on 7–50% linear, low-$Mg^{2+}$ sucrose gradients prepared in the same buffer and run at 260,800 × g in a SW41 Ti rotor for 4.5 h at 4 °C. After centrifugation, fractions of 0.5 ml were collected manually; these were then adjusted to a final concentration of 10 mM Tris-HCl, pH 7.5, 10 mM EDTA, and 0.5% SDS. Total RNA was isolated by two consecutive extractions with 10 mM Tris-HCl, pH 7.5 saturated phenol:chloroform:isoamyl alcohol (25:24:1), followed by a chloroform:isoamyl alcohol (24:1) extraction. Total RNA was precipitated with ethanol in the presence of 0.3 M sodium acetate, pH 5.2. Finally, RNA pellets were dissolved in 20 μl of distilled water and 10 μl were resolved on a 7% polyacrylamide, 8 M urea gel and subjected to northern blotting using specific [32P]-labelled oligonucleotides as probes (Supplementary Table 4).

**RNA extraction and northern blotting experiments**. Extraction of yeast total RNA was performed as follows[12]: dry yeast cell pellets were resuspended with 0.5 ml water-saturated phenol and 0.5 ml guanidine thiocyanate (GTC) mix (50 mM Tris-HCl, pH 8.0, 10 mM EDTA, pH 8.0, 4 M guanidine thiocyanate, 2% N-Lauroylsarcosine, 143 mM β-Mercaptoethanol). Cells were broken by vigorous vortexing three times for 2 min at 4 °C in the presence of glass beads. The resulting samples were mixed with 7.5 ml water-saturated phenol and 7.5 ml GTC mix and incubated for 5 min at 65 °C. After addition of 7.5 ml chloroform and 4 ml sodium acetate buffer (10 mM Tris-HCl, pH 8.0, 1 mM EDTA, pH 8.0, 100 mM sodium acetate), samples were mixed vigorously and centrifuged at 3220 × g for 5 min at 4 °C. Aqueous phases were recovered and RNAs were re-extracted two additional times with water-saturated phenol:chloroform (1:1). RNAs were then concentrated by ethanol precipitation and ultimately resuspended in ultrapure $H_2O$. In all northern blotting experiments (see below), equal amounts of these total RNAs (4 μg) were analyzed.

For northern blotting analyses of high-molecular-mass RNA species, RNAs were separated as described in "Molecular Cloning", Sambrook and Russell, CSHL Press ("Separation of RNA According to Size: Electrophoresis of Glyoxylated RNA through Agarose Gels"). RNAs were then transferred to Hybond N + membranes (GE Healthcare) by capillarity using 5× SSC as a transfer buffer. Low-molecular-mass RNA species were separated by electrophoresis through 6% acrylamide:bisacrylamide (19:1), 8 M urea gels using 1× TBE as a running buffer. RNAs were then transferred to Hybond N membranes (GE Healthcare) by electro-transfer in 0.5× TBE buffer, 20 V, 4 °C, overnight. In all cases, membranes were hybridized with 32P-labelled oligonucleotide probes using Rapid-hyb buffer (GE Healthcare). Radioactive membranes were exposed to Phosphorimaging screens and revealed using Typhoon TRIO or Typhoon 9400 Variable Mode Imagers (GE Healthcare) driven by Typhoon Scanner Control software (Version 5.0). Sequences of the oligonucleotides used as probes in this study are described in the Supplementary Table 4. Quantifications of the signals were performed using PhosphorImager data and Multi Gauge software (Version 3.0, FUJIFILM). Statistically significant differences were determined using one-tailed Student's t-test (https://www.socscistatistics.com/tests/studentttest/default2.aspx).

**Western blotting experiments**. For western blotting experiments, protein samples were separated on 10% SDS-polyacrylamide gels and transferred to nitrocellulose membranes using a Trans-Blot Turbo apparatus (BioRad). Membranes were saturated for 1 h with PBST buffer (137 mM NaCl, 2.7 mM KCl, 10 mM $Na_2HPO_4$, 2 mM $KH_2PO_4$, 0.1% Tween-20) containing 5% (w/v) powder milk. Following incubation for 2 h with the same buffer containing the primary antibodies, membranes were rinsed three times for 5 min with PBST buffer, incubated for 1 h with the secondary antibodies diluted in PBST containing 5% (w/v) powder milk and finally washed three times for 10 min with PBST buffer. Luminescent signals were generated using the Clarity Western ECL Substrate (Bio-Rad), captured using a ChemiDoc Touch Imaging System (Bio-Rad) and quantified using the Image Lab software (Version 5.2.1, Bio-Rad). HA-tagged proteins were detected

using HRP-conjugated mouse monoclonal anti-HA antibodies (Roche Diagnostics, Cat. # 12013819001, 1:1000 dilution); Pgk1 protein was detected using mouse monoclonal anti-Pgk1 antibodies (Clone 1086CT10.2.1, Invitrogen, Cat. # MA5-37712, 1:8000 dilution). Primary antibodies used to detect Npa1 (1:5000 dilution), Dbp6 (1:10,000 dilution), Nop8 (1:2000 dilution), Rsa3 (1:10,000 dilution), Nhp2 (1:5000 dilution), and Prp43 (1:4000 dilution) were generated by custom antibody production services and described elsewhere[25,55,56]. Secondary antibodies were purchased from Promega (HRP-conjugated anti-mouse antibodies, Cat. # W402B, 1:10,000 dilution; HRP-conjugated anti-rabbit antibodies, Cat. # W401B, 1:10,000 dilution).

**Immunoprecipitation experiments**. Cell pellets corresponding to 500 ml cultures at $OD_{600}$ ~0.6 were resuspended with approximately one volume of ice-cold A200-KCl buffer [20 mM Tris-HCl, pH 8.0, 5 mM magnesium acetate, 200 mM KCl, 0.2% Triton-X100] supplemented with 1 mM DTT, 1× Complete EDTA-free protease inhibitor cocktail (Roche), 0.1 U/μl RNasin (Promega). About 400 μl of ice-cold glass beads (0.5 mm diameter, Sigma) were added to 800 μl aliquots of the resuspended cells, which were broken by vigorous shaking, two times 30 sec separated by 1 min incubation on ice using Precellys (Ozyme). Extracts were clarified through two successive centrifugations at 16,000 × g and 4 °C for 5 min and quantified by measuring $A_{260}$. Equal amounts of soluble extracts were incubated for 2 h at 4 °C with 15 μl (bed volume) of immunoglobulin G (IgG)-Sepharose beads (GE Healthcare) in a total volume of 1 ml (adjusted to A200-KCl buffer supplemented with 1 mM DTT, 1× Complete EDTA-free protease inhibitor cocktail, 0.1 U/μl RNasin) on a rocking table. For RNA analyses, beads were washed seven times with 1 ml of ice-cold A200-KCl buffer supplemented with 1 mM DTT and 1× Complete EDTA-free protease inhibitor cocktail. RNA was extracted from bead pellets as follows: 160 μl of 4 M guanidium isothiocyanate solution, 4 μl of glycogen (Roche), 80 μl of [10 mM Tris-HCl, pH 8.0, 1 mM EDTA, pH 8.0, 100 mM sodium acetate], 120 μl of phenol and 120 μl of chloroform were added. Tubes were shaken vigorously, incubated 5 min at 65 °C, and centrifuged 5 min at 16,000 × g (4 °C). Aqueous phases (240 μl) were mixed vigorously with 120 μl of phenol and 120 μl of chloroform, centrifuged 5 min at 16,000 × g (4 °C) and the resulting aqueous phases were ethanol precipitated. For protein analyses, beads were washed five times with 1 ml of ice-cold A200-KCl buffer supplemented with 1 mM DTT and 1× Complete EDTA-free protease inhibitor cocktail and two additional times with 1 ml of A200-NaCl buffer [20 mM Tris-HCl, pH 8.0, 5 mM magnesium acetate, 200 mM NaCl, 0.2% Triton-X100] supplemented with 1 mM DTT, 1× Complete EDTA-free protease inhibitor cocktail. Beads were resuspended directly with 2× SDS gel-loading buffer [100 mM Tris-HCl, pH 6.8, 4% SDS, 20% glycerol, 200 mM DTT, 0.2% bromophenol blue].

**Purification of recombinant proteins from *E. coli***. The coding sequence of *DBP7* was cloned into a pQE80-derived plasmid for expression of N-terminally His$_{10}$-ZZ-tagged recombinant proteins in *E. coli* (His$_{10}$-ZZ-Dbp7$_{WT}$). Site-directed mutagenesis was used to introduce mutations leading to substitution of lysine 197 for alanine (His$_{10}$-ZZ-Dbp7$_{K197A}$). Plasmids were used to transform BL21 Codon Plus *E. coli* and protein expression was induced with 1 mM ITPG overnight at 18 °C. After harvesting, cell pellets were resuspended in a lysis buffer containing 50 mM Tris-HCl, pH 7.0, 500 mM NaCl, 1 mM $MgCl_2$ 10 mM imidazole, 1 mM PMSF, 10% glycerol. Cells were lysed by sonication and the lysate cleared by centrifugation. Polyethyleneimine was added to a final concentration of 0.05% and precipitated nucleic acids were removed. The lysate was incubated with cOmplete™ His-Tag purification resin (Roche) and after binding, the matrix was washed using buffers containing (i) 50 mM Tris-HCl, pH 7.1, 500 mM NaCl, 1 mM $MgCl_2$, 30 mM imidazole, 10% glycerol, and (ii) 50 mM Tris-HCl, pH 7.1, 1 M NaCl, 1 mM $MgCl_2$, 30 mM imidazole, 10% glycerol. Elution of bound proteins were performed using a buffer containing 50 mM Tris-HCl, pH 7.0, 500 mM NaCl, 1 mM $MgCl_2$, 300 mM imidazole and 10% glycerol. Purified protein was dialyzed against a buffer containing 50 mM Tris-HCl, pH 7.0, 120 mM NaCl, 2 mM $MgCl_2$, 20% glycerol. A Bradford assay was used to determine protein concentration.

**In vitro NADH-coupled ATPase assay**. In vitro NADH-coupled ATPase assays[17,19] were performed to monitor ATP hydrolysis by purified His$_{10}$-ZZ-Dbp7$_{WT/K197A}$. Each reaction contained 50 mM Tris-HCl, pH 7.4, 25 mM NaCl, 2 mM $MgCl_2$, 4 mM ATP, 1 mM phosphoenolpyruvate, 300 μM NADH, and 20 U/ml pyruvate kinase/lactic dehydrogenase. Control reactions containing no protein were performed to establish a background rate of ATP hydrolysis that was subtracted from all other samples. Samples contained 1.5 μM purified His$_{10}$-ZZ-Dbp7$_{WT}$ or His$_{10}$-ZZ-Dbp7$_{K197A}$ and 0–4 μM RNA (5′- UUUUUUUUUUUUUUUUUUUUUUUUUUUUUUUU-3′). The $A_{340}$ was measured every 50 sec for 30 min at 30 °C using a Gen5 Microplate Reader (Biotek). The amount of ATP hydrolysed, equimolar to the amount of NADH oxidized, was determined from the slope of the linear absorbance decrease using the following equation in which $K_{path}$ is the molar absorption co-efficient of the optical path:

$$\text{nmol ATP hydolysed } x \sec^{-1} = \frac{dA_{340}}{dt} \times K_{path}^{-1} \times 10^6$$

**Reporting summary**. Further information on research design is available in the Nature Research Reporting Summary linked to this article.

## Data availability

All the source data used for this study (raw images, graphs, and quantifications) are provided in the "Source Data file" available in the supplementary material. Other data are available from the corresponding author upon request. Source data are provided with this paper.

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

## Acknowledgements

Our work has benefited from fruitful technical and scientific contacts with the neighboring groups of Jérôme Cavaillé (J.C., Jade Hebras), Olivier Gadal (O.G., Christophe Dez, Frédéric Beckouët), Pierre-Emmanuel Gleizes (Simon Lebaron), and Tamás Kiss at the LBME/CBI and Alain Kamgoué. We thank Jade Hebras for technical help with the detection of ribose-methylated nucleosides. We thank Hussein Hamze for experiments related to the study not presented in this manuscript. We thank all members of the Henry/Henras team for helpful discussions. The Henry/Henras group is supported by grants from ANR (ANR-20-CE12-0026) and funding from CNRS and University of Toulouse. R.A.M. is supported by grants from the *Rectorat* of Lebanese University. M.J. is supported by a Ph.D. fellowship from the Lebanese University and CIOES Organization. The group of J.d.l.C. is supported by the Spanish Ministry of Science and Innovation [PID2019-103859-GB-I00 AEI/10.13039/501100011033], and the Andalusian Regional Government (JA; BIO-271). J.C. was supported by a Ph.D. fellowship (PIF) from the University of Seville, and S.M.-V. is an academic research staff of the JA (PAIDI2020). M.T.B. and K.E.B. are supported by funding from the Deutsche Forschungsgemeinschaft (SFB860) and the University Medical Centre Göttingen. We thank F. Espinar-Marchena and H. Domínguez-Martín for assistance during the initial design of the rRNA suppressor screen.

## Author contributions

M.J., J.C., P.V., O.H., N.J.W., E.V., M.T.B., K.E.B., Y.H., R.A.M., J.d.l.C. and A.H. designed the experiments. Experiments were performed by M.J., J.C., C.D., S.M.-V., R.C., P.V., O.R.-G., C.V., Y.H., E.V., M.T.B., K.E.B., J.d.l.C. and A.H. All authors interpreted the data. M.J., J.d.l.C. and A.H. wrote the manuscript with strong contributions from Y.H.

## Competing interests

The authors declare no competing interests.
