## [Peer Review File · Nature Communications]

Association of snR190 snoRNA chaperone with early pre-60S particles is regulated by the RNA helicase Dbp7 in yeastREVIEWER COMMENTS

Reviewer #1 (Remarks to the Author):

In this manuscript Jaafar et al. describe a series of elegant experiments that reveal Dbp7 as the enzyme responsible for removal of snR190. The authors combine a clever genetic screen to identify ribosomal RNA suppressor mutations that can alleviate the absence of Dbp7. Following the identification of an rRNA mutation that presumably weakens the base pairing interactions with snR190, the authors show that snR190 is an rRNA chaperone and not a targeting snoRNA for methylation of G2395. The interplay between Dbp7 and snR190 is further illustrated by the observations that snR190 accumulates in early pre-60S particles and that the depletion of snR190 can alleviate the deletion of Dbp7.

In the opinion of this reviewer, this is an excellent manuscript that is very polished. The clarity of the presented data and its key relevance for ribosome assembly as well as RNA biology mean that this paper is of immediate interest to the broad readership of Nature Communications and should be published as soon as the following very minor comments have been addressed.

1. The authors use the abbreviation AFM for assembly and maturation factor. Here the commonly used AF (for assembly factor) would be sufficient as to reduce the number of commonly used acronyms.
2. The number of panels in figures 1 and 3 could be reduced. For example, figure 1 could be restructured somewhat so that only the most essential data is presented. This would further enhance the clarity of the presented data. Specifically, Figure 1a and b could be supplementary information. For figure 3 panel A could be provided as supplementary information.

Sebastian Klinge

Reviewer #2 (Remarks to the Author):

Ribosome biogenesis requires many transiently interacting proteins and snoRNAs which coordinate the assembly process and rRNA processing in a spatiotemporal manner. The knowledge about the detailed function of many snoRNAs is sparse. The authors used a genetic screen in yeast to isolate suppressors of phenotypes of a DBP7 deletion strain. DBP7 codes for a non-essential DEAD-box protein, which has been previously reported to be involved in the synthesis of the large ribosomal subunit (LSU). One of the suppressor mutants had a point mutation in helix 73 of domain V of the LSU RNA. Interestingly, the point mutation lies within the sequence hybridizing with snoRNA snR190 in the pre-ribosome. To investigate if Dbp7 has an snR190 related function, the authors performed a thorough analysis if and how snR190 may affect pre-60S subunit synthesis. The authors show that snR190 supports proper pre-60S biogenesis and confirm that snR190 is not required for methylation of the ribose at position G2395, as previously predicted. Mutagenesis of the two different snR190-binding sites in domains V and I respectively, revealed that pre-60S biogenesis is only affected if both binding sites have been altered. This led to the assumption that snR190 acts as a chaperone which either bridges the two domains during ribosome maturation or prevents premature interactions of the two domains with other parts of the nascent ribosome. Based on further genetic and biochemical studies it is suggested that snR190 mediates the association of the Npa1 complex to early pre-60S subunits and that Dbp7 is required for dissociation of snR190 and other snoRNAs as well as members of the Npa1 complex from pre-60S subunits.

The study is the outcome of carefully performed experiments revealing a yet unknown function of snR190 in biogenesis of the large ribosomal subunit. However, the impact of snR190 on proper pre-60S maturation remains partly undefined. For instance, snR190 inactivation results in a clear delay in RNA processing, but why it interferes with ongoing ribosome maturation and how crucial it is for ongoing ribosome maturation could be better pointed out.

Problems

- 1) According to Fig. 1A, suppressor 10 which contains the C2392U mutation efficiently rescued the

growth defect observed when Dbp7 was depleted. In contrast, all snR190 mutations analyzed were only weak suppressors of the growth defect resulting from Dbp7 depletion. It is therefore questionable that impaired snR190 binding to its cognate rRNA domains provokes the strong growth recovery.

2) Both isolated suppressor mutants contained mutations in addition to the C2392U mutation. It is possible that full suppression is due to a cumulative effect of several mutations. Reverting the C2392U mutation should clarify whether specifically this mutation is responsible for full suppression of the growth defect. It would be helpful to indicate the positions of the other mutations.

3) Ribosome synthesis is essential for growth. Depletion of either Dbp7 or mutagenesis of snR190 resulted in comparable pre-rRNA processing defects (27SB/27SA2 ratio), whereas cellular growth was clearly differently affected. If impaired pre-rRNA processing is responsible for the observed growth reduction the growth defects should be similar in the different mutants. Please comment!

4) No evidence is provided that the C2392U mutation is directly linked to snR190 dissociation. This should be analyzed.

5) The authors argue that C2392U mutation might lead to snR190 dissociation from helix 73. On the other hand, mutations in BoxA of snR190 should also impair its association with helix 73. Do both mutants have a similar pre-rRNA processing phenotype in the absence and presence of Dbp7? This should be tested.

Minor points

1) Please quantify the 27SB/27SA2 ratio in snR190 mutants (Fig. 2 and Fig. 4B) in comparison to the respective ratio in $\Delta dbp7$ (Fig. 6).

2) Depletion of Dbp7 results in 7S and 5.8S rRNA reduction. Has such a reduction be observed in snR190 mutants?

3) Generation times presented in supplementary table 2 could be presented as bar chart to better visualize the differences between $\Delta dbp7$ and snR190 mutants. Please explain why the doubling time is different in W303 snr190-[mut.C] and W303 snr190-[mut.C] + E.V..

4) To investigate whether bridging of snR190 between domain V and domain I is crucial for pre-60S maturation, BoxA and BoxB containing snR190 fragments could be independently expressed in yeast cells. In those cells, association of the Npa1 complex with pre-60S particles could be tested and pre-rRNA processing could be analyzed.

5) Fig. 7A. It should be better explained why depletion of Npa2 was used to study interaction between the Npa1-complex and snR190. Since the authors don't show a direct interaction between Npa1 and snR190, they should not put too much emphasis on the fact that both components might physically interact.

6) Fig. 6C, second bar: wrong labelling $\Delta DBP7$ snR190 WT instead of DBP7 WT mut.C

Reviewer #3 (Remarks to the Author):

Many helicases are involved in ribosome biogenesis, however little is known about mechanisms involved. In this study, Jaafar et al (NCOMMS-20-49545) show an interplay between Dbp7 helicase and snR190 snoRNA during early 60S particle biogenesis. They provide evidence that yeast snR190 does not function as a methylation guide but acts as an RNA chaperone of 25S rRNA in cooperation with Npa1 complex. The described data are novel and convincing, supported by high quality complementary genetic and biochemical experiments. I find the paper very interesting, it merit publication.

Points that should be addressed:

1) 2 plasmids encoding mutant 35S pre-rRNA were shown to improve growth of a strain depleted for Dbp7. Besides several mutations, the authors point the C2392U mutation in 25S rRNA as responsible

for the phenotype. That is probably true, but did the authors tried to reverse only this mutation in the two plasmids and observed growth defect again ?

2)The authors should comment on growth inhibition of sup10 strain compared to the control on SGal medium (fig 1A) ? also on the Dbp7 higher expression in this mutant (fig 1B) if it exist? The coomassie staining seems to indicate equal protein amounts, but an internal control as pgk1 was measured ?

3)Page 11 the authors indicate that loss of function of snR190 leads to defects on maturation pathway of 25S and 5.8S. This is only shown for 25 S, not for 5.8S ?

4)Figure 5C : can the authors comment on snR42 which is also displaced in dbp7 mutant ? on snR5 large decrease in dbp7 mutant ?

5)Was it shown that the Dbp7 K197A ATPase activity is null ?

6)Fig 6A: The lack of snR190 alleviates the growth defect of a dbp7 null mutant, is that true also for the dbp7 K197A mutant ?

7)The pre-rRNA processing defects observed in the absence of snR190 are present but weak. Authors should be careful since differences are observed between different clones (see Figure 2A, level of 25S in WT 1 and 2, level of 25S between snr190 mut C 1 and 2; also in Supp Fig3, in Fig 4B, 27SA2 and 27SB in mutAB 1 and 2 are very different). The authors should define more clearly the error bars. 2 clones were used but how many experiments were done with each?

8)Fig 3B: a WT strain could have been shown on the same figure

Minor points , typos

1)Fig 2A: What is the difference between strains W303 and WT1, WT2

2)Supp Table 2 : there is a difference between doubling time of W303 snR190mutC and the same strain with empty vector which renders the interpretation of complementation difficult ? This is not observed in BY4741 background

3)Supp Fig 2: the pink color used both for snoRNA and 25S brings confusion

4)Supp Fig 4B row 7 and 8 : write mutB instead of mutA

5)Results p7: the genetic link is shown between Dbp7 and 25S rRNA, not directly with snR190

6)Figure 6A: position of pre60S and 90S should be indicated

REVIEWER COMMENTS

Reviewer #1 (Remarks to the Author):

In this manuscript Jaafar et al. describe a series of elegant experiments that reveal Dbp7 as the enzyme responsible for removal of snR190. The authors combine a clever genetic screen to identify ribosomal RNA suppressor mutations that can alleviate the absence of Dbp7. Following the identification of an rRNA mutation that presumably weakens the base pairing interactions with snR190, the authors show that snR190 is an rRNA chaperone and not a targeting snoRNA for methylation of G2395. The interplay between Dbp7 and snR190 is further illustrated by the observations that snR190 accumulates in early pre-60S particles and that the depletion of snR190 can alleviate the deletion of Dbp7.

In the opinion of this reviewer, this is an excellent manuscript that is very polished. The clarity of the presented data and its key relevance for ribosome assembly as well as RNA biology mean that this paper is of immediate interest to the broad readership of Nature Communications and should be published as soon as the following very minor comments have been addressed.

1. The authors use the abbreviation AFM for assembly and maturation factor. Here the commonly used AF (for assembly factor) would be sufficient as to reduce the number of commonly used acronyms.

We are deeply grateful to Reviewer #1 for the very positive evaluation of our study. We changed the "AMF" abbreviation to "AF" throughout the manuscript.

2. The number of panels in figures 1 and 3 could be reduced. For example, figure 1 could be restructured somewhat so that only the most essential data is presented. This would further enhance the clarity of the presented data. Specifically, Figure 1a and b could be supplementary information. For figure 3 panel A could be provided as supplementary information.

Figure 1 had to be further modified according to the requests of Reviewers #2 and #3. To follow the advice of Reviewer #1, we have moved panel (b) to the supplementary material (Supplementary Fig. 3 in the revised manuscript). However, we would propose to leave panel (a) and the new data requested by Reviewers #2 and #3 (new panel b) in the main figure because these are actually the most important data of the figure, showing that growth of the *dbp7Δ* strain is rescued by the C2392U mutation in 25S rRNA.

In agreement with the advice of Reviewer #1, panel (a) of Figure 3 has been moved to the supplementary material and is now presented in Supplementary Fig. 9 of the revised manuscript.

Reviewer #2 (Remarks to the Author):

Ribosome biogenesis requires many transiently interacting proteins and snoRNAs which coordinate the assembly process and rRNA processing in a spatiotemporal manner. The knowledge about the detailed function of many snoRNAs is sparse. The authors used a genetic screen in yeast to isolate suppressors of phenotypes of a DBP7 deletion strain. DBP7 codes for a non-essential DEAD-box protein, which has been previously reported to be involved in the synthesis of the large ribosomal subunit (LSU). One of the suppressor mutants had a point mutation in helix 73 of domain V of the LSU RNA. Interestingly, the point mutation lies within the sequence hybridizing with snoRNA snR190 in the pre-ribosome. To investigate if Dbp7 has an snR190 related function, the authors performed a thorough analysis if and how snR190 may affect pre-60S subunit synthesis. The authors show that snR190 supports proper pre-60S biogenesis and confirm that snR190 is not required for methylation of the ribose at position G2395, as previously predicted. Mutagenesis of the two different snR190-binding sites in domains V and I respectively, revealed that pre-60S biogenesis is only affected if both binding sites have been altered. This led to the assumption that snR190 acts as a chaperone which either bridges the two domains during ribosome maturation or prevents premature interactions of the two domains with other parts of the nascent ribosome. Based on further genetic and biochemical studies it is suggested that snR190 mediates the association of the Npa1 complex to early pre-60S subunits and that Dpb7 is required for dissociation of snR190 and other snoRNAs as well as members of the Npa1 complex from pre-60S subunits.

The study is the outcome of carefully performed experiments revealing a yet unknown function of snR190 in biogenesis of the large ribosomal subunit. However, the impact of snR190 on proper pre-60S maturation remains partly undefined. For instance, snR190 inactivation results in a clear delay in RNA processing, but why it interferes with ongoing ribosome maturation and how crucial it is for ongoing ribosome maturation could be better pointed out.

Problems

1) According to Fig. 1A, suppressor 10 which contains the C2392U mutation efficiently rescued the growth defect observed when Dbp7 was depleted. In contrast, all snR190 mutations analyzed were only weak suppressors of the growth defect resulting from Dbp7 depletion. It is therefore questionable that impaired snR190 binding to its cognate rRNA domains provokes the strong growth recovery.

The genetic screen described in Fig. 1A was performed using the strain system constructed by Nomura's lab, which contains only one active copy of the rDNA on a multi-copy plasmid. Indeed, given the repetitive nature of rDNA genes, this genetic trick was the only possible approach allowing to identify rRNA suppressor mutations that alleviate the absence of Dbp7. However, this strain grows very poorly and displays nucleoli that are different from those of a "natural" strain expressing rRNAs from the genomic locus, indicating that ribosome biogenesis is altered to some extent. Reviewer #2 raises the point that in this genetic background, the 25S rRNA C2392U mutation efficiently rescues the growth defect due to Dbp7 depletion. However, this conclusion cannot be drawn from the data we showed in the original manuscript, because the growth assays we showed were comparing the growth rate of the EMY65 strain (*GAL::3HA::DBP7*), with or without the suppressor mutation, on galactose (Dbp7 expressed) and glucose (Dbp7 repressed). We clarified this point in the revised manuscript, showing the

appropriate data to address this comment of Reviewer #2. We now show in Fig. 1b a comparison of the growth rate of strain EMY65 (*GAL::3HA::DBP7*) expressing (Sup. #10) or not (pNOY373) the suppressor mutation with the growth of the “wild-type” Nomura strain (Dbp7 expressed from its endogenous promoter) on a glucose-containing medium. This comparison clearly shows that the C2392U mutation does not fully rescue the growth defect due to Dbp7 depletion in this genetic background, similar to what we observed in absence of snR190 in the BY4741/W303 backgrounds. It is however difficult to draw more precise conclusions from these data since, given the strong growth defect of the Nomura’s strain, ribosome biogenesis cannot be identical to that of the standard BY4741 and W303 strains in which we mutated snR190. Many parameters likely differ between the Nomura’s and the BY4741 and W303 strains such as the snoRNA accessibility to their rRNA sites, the kinetics of pre-rRNA processing, the number of aberrant pre-ribosomal particles, etc. The requirement for Dbp7/snR190 functions may be different depending on the genetic background, leading to a lesser suppression of the *dbp7Δ* growth phenotype by snR190 mutations in the standard BY4741 and W303 strains. These new data have been included in Fig. 1b of the revised manuscript and described in the main text (page 6, line 26).

2) Both isolated suppressor mutants contained mutations in addition to the C2392U mutation. It is possible that full suppression is due to a cumulative effect of several mutations. Reverting the C2392U mutation should clarify whether specifically this mutation is responsible for full suppression of the growth defect. It would be helpful to indicate the positions of the other mutations.

We fully agree with this comment. To remove any ambiguity, we introduced the C2392T mutation in the wild-type pNOY373 plasmid to generate a mutant plasmid bearing as a sole mutation the C2392T mutation (pNOY373 C2392T in current Fig. 1a). In parallel, we also reverted solely the T2392 mutation to the wild-type C2392 nucleoside in the Sup. #2 and Sup. #10 suppressor plasmids (Sup. #2 T2392C and Sup. #10 T2392C in current Fig. 1a), leaving unchanged all the other mutations. These plasmids were transformed into the original conditional *dbp7* strain generated in the Nomura’s strain and the growth of the resulting strains was monitored at 30 °C on selective SD medium. These experiments allowed us to show unambiguously that the C2392U mutation in the 25S rRNA was by itself responsible for the suppression of the growth defect of the *dbp7Δ* strain. As indicated above, these important data have now been included in Fig. 1a of the revised version of the manuscript and described in the main text (page 6, line 21).

3) Ribosome synthesis is essential for growth. Depletion of either Dbp7 or mutagenesis of snR190 resulted in comparable pre-rRNA processing defects (27SB/27SA2 ratio), whereas cellular growth was clearly differently affected. If impaired pre-rRNA processing is responsible for the observed growth reduction the growth defects should be similar in the different mutants. Please comment!

Although depletion of Dbp7 or mutagenesis of snR190 indeed result in comparable pre-rRNA processing defects qualitatively, the effects are more severe upon Dbp7 depletion, which is consistent with the more severe growth defect of the *dbp7Δ* strain compared to the strain lacking snR190. The ribosomal subunit profiles presented in the original manuscript (Fig. 2, 4 and 5 of the revised manuscript) also showed that ribosome biogenesis and translation defects are more severe in absence of Dbp7 than in absence of snR190. Comparison of the polysome profiles revealed a stronger reduction of the free 60S subunit and more prominent half-mer

polysomes. In the discussion of the original manuscript, we had mentioned the following: “*The growth and pre-rRNA processing defects resulting from lack of snR190 are less severe than those observed in the absence of Dbp7*”, and we then discussed the possible reasons. To make this result clearer in the revised manuscript, we performed additional Northern blotting experiments to show all the samples together on the same figure. These additional data show that the 27SB/27SA₂ ratio is more severely decreased in absence of Dbp7 than in absence of snR190. These data are presented in Supplementary Fig. 6 of the revised manuscript and referred to in the discussion section (page 18, line 20). We also took the opportunity of this comment of Reviewer #2 to develop more the discussion on the potential other functions of Dbp7 (page 18, Line 21).

4) No evidence is provided that the C2392U mutation is directly linked to snR190 dissociation. This should be analyzed.

We agree with reviewer #2 that we did not formally prove that the C2392U mutation was directly linked to snR190 dissociation. The reasons behind this was that the only way to address this question was to use the yeast strain designed for the genetic screen in which all rRNAs are expressed from a multicopy plasmid. As mentioned above (point #1 of Reviewer #2), this strain is very sick and we did not feel comfortable to perform functional studies with this strain. However, during the revision period we analyzed the sedimentation profile of snR190 and snR42 in strain EMY65 originally used for the genetic screen transformed with either the pNOY373 plasmid bearing the specific C2392T mutation or the Sup#10 plasmid with the reverted mutation (T2392C). We observed a slight but reproducible change in the sedimentation profile of snR190 in these strains consistent with the model that the C2392U mutation partially alleviates retention of snR190 into pre-ribosomes. These data are presented in Supplementary Fig. 13 of the revised manuscript and referred to in the discussion section (page 16, Line 5).

5) The authors argue that C2392U mutation might lead to snR190 dissociation from helix 73. On the other hand, mutations in BoxA of snR190 should also impair its association with helix 73. Do both mutants have a similar pre-rRNA processing phenotype in the absence and presence of Dbp7? This should be tested.

Our data presented in Fig. 4 show that mutation of box A of snR190 on its own (in the presence of Dbp7) slightly affects pre-rRNA processing (Fig. 4b and Supplementary Fig. 8) and does not induce a significant growth defect (Supplementary Table 2 and Supplementary Fig. 15). In absence of Dbp7, our data presented in the original manuscript show that mutation of box A of snR190 partially rescues the pre-rRNA processing (now in Fig. 7b, c of the revised manuscript) and growth defects in the conventional BY4741 and W303 genetic backgrounds (Supplementary Table 2 and Supplementary Fig. 15). We now show in the revised version of the manuscript that the C2392T mutation also partially suppresses the pre-rRNA processing and growth defects of the EMY65 strain expressing rRNAs from a single rDNA copy harboured in a multicopy plasmid. Therefore, as requested by Reviewer #2, we now show in the revised manuscript that mutation C2392U in the 25S rRNAs and mutation of snR190 box A have similar consequences on growth and pre-rRNA processing in absence of Dbp7. These results are presented in Supplementary Fig. 14 of the revised manuscript and described in the main discussion section (page 16, line 10).

Minor points

1) Please quantify the 27SB/27SA₂ ratio in snR190 mutants (Fig. 2 and Fig. 4B) in comparison to the respective ratio in Δ dbp7 (Fig. 6).

This point is related to major point 3 of Reviewer #2. We have quantified the 27SB/27SA₂ ratios in Fig. 2a (W303) and Supplementary Fig. 4a (BY4741), which are now shown in Supplementary Fig. 5c, d of the revised manuscript. We had already quantified the 27SB/27SA₂ ratios in Fig. 4b (W303), which were shown along with those in the BY4741 background in Supplementary Fig. 4 of the original manuscript, now in Supplementary Fig. 8 of the revised manuscript. We further added to the revised manuscript a supplementary figure showing Northern blot data with all the samples on the same figure and the corresponding quantifications (Supplementary Fig. 6).

2) Depletion of Dbp7 results in 7S and 5.8S rRNA reduction. Has such a reduction been observed in snR190 mutants?

We analysed by Northern blotting the accumulation levels of the 7S pre-rRNA precursor to the mature 5.8S rRNA in absence of snR190. We did not detect a significant change in the steady state levels of the 7S pre-rRNA, suggesting that the defect in the production of the 27SB intermediates in absence of snR190 is not strong enough to have repercussions on the late maturation steps of 5.8S rRNAs (Supplementary Fig. 6). In agreement with this, analysis of the steady-state levels of the mature 5.8S rRNAs in the different strains did not vary either (Supplementary Fig. 7). These data are described in the revised manuscript page 8, line 20.

3) Generation times presented in supplementary table 2 could be presented as bar chart to better visualize the differences between Δ dbp7 and snR190 mutants. Please explain why the doubling time is different in W303 snr190-[mut.C] and W303 snr190-[mut.C] + E.V.

In addition to Supplementary Table 2, we added a bar chart to better visualize the differences between all strains (Supplementary Fig. 15).

Concerning the surprising difference in growth rate of the W303 snr190-[mut.C] and W303 snr190-[mut.C] + E.V. strains, we thank Reviewers #2 and #3 (Minor point #2) for having noticed this inconsistency that we missed in the original submission. We repeated the experiments with all the strains in the W303 background and calculated the corresponding doubling times. For reasons that we retrospectively do not understand (e.g. media problems), the doubling time of the W303 snr190-[Mut.C] with the empty vector was not correct. We now provide the accurate values in Supplementary Table 2 and Supplementary Fig. 15 of the revised manuscript.

4) To investigate whether bridging of snR190 between domain V and domain I is crucial for pre-60S maturation, BoxA and BoxB containing snR190 fragments could be independently expressed in yeast cells. In those cells, association of the Npa1 complex with pre-60S particles could be tested and pre-rRNA processing could be analyzed.

We understand this point of Reviewer #2 but we did not engage into complex constructions and experiments to address this point for the following reason. Our results presented in the original manuscript show that mutation of either box A or box B does not drastically impact pre-rRNA processing and growth. Mutation of both boxes is required to induce defects comparable to those observed in the absence of snR190. We concluded from these data that snR190 may not function in the bridging of domains I and V, but rather in the shielding of its complementary sequences to prevent them from base-pairing inappropriately. On this basis, we doubted that expressing independently sub-fragments of snR190 containing either box A and box B would actually change the association of the Npa1 complex or induce pre-rRNA processing defects.

5) Fig. 7A. It should be better explained why depletion of Npa2 was used to study interaction between the Npa1-complex and snR190. Since the authors don't show a direct interaction between Npa1 and snR190, they should not put too much emphasis on the fact that both components might physically interact.

We agree with Reviewer #2 that the rationale behind this experiment had not been explained sufficiently in the original manuscript. We also agree that the conclusions of this experiment were limited. Given that we had to shorten substantially the length of the manuscript to fit with the format of *Nature Communications*, we decided to remove this panel and its description in the revised manuscript.

6) Fig. 6C, second bar: wrong labelling Δ DBP7 snR190 WT instead of DBP7 WT mut.C We thank Reviewer #2 for this observation and we apologise for the mistake. The labelling of the figure has been corrected in the revised manuscript (now Fig. 7c).

Reviewer #3 (Remarks to the Author):

Many helicases are involved in ribosome biogenesis, however little is known about mechanisms involved. In this study, Jaafar et al (NCOMMS-20-49545) show an interplay between Dbp7 helicase and snR190 snoRNA during early 60S particle biogenesis. They provide evidence that yeast snR190 does not function as a methylation guide but acts as an RNA chaperone of 25S rRNA in cooperation with Npa1 complex. The described data are novel and convincing, supported by high quality complementary genetic and biochemical experiments. I find the paper very interesting, it merit publication.

We are sincerely grateful to Reviewer #3 for this very positive comment.

Points that should be addressed:

1) 2 plasmids encoding mutant 35S pre-rRNA were shown to improve growth of a strain depleted for Dbp7. Besides several mutations, the authors point the C2392U mutation in 25S rRNA as responsible for the phenotype. That is probably true, but did the authors tried to reverse only this mutation in the two plasmids and observed growth defect again?

This very relevant point has also been raised by Reviewer #2 (major point 2). As described above, in response to Reviewer #2, we have both reversed the C2392T mutation in suppressor plasmids Sup. #2 and Sup. #10 and introduced, specifically and exclusively, the C2392T mutation in pNOY373. These plasmids were transformed into the EMY65 strain and the analysis of the phenotypes confirmed that 25S rRNA mutation C2392U is both necessary and sufficient to suppress the growth defect of the *dbp7Δ* strain. These data have been added to the revised version of the manuscript (Fig. 1a) and described in the main text (page 6, line 21).

2) The authors should comment on growth inhibition of sup10 strain compared to the control on SGal medium (fig 1A)? also on the Dbp7 higher expression in this mutant (fig 1B) if it exist? The coomassie staining seems to indicate equal protein amounts, but an internal control as pgk1 was measured?

We thank the reviewer for pointing out this peculiar phenomenon. In fact, we repeated several times this experiment during the revision period, but failed to reproduce this observation. We now show in Fig. 1a of the revised version of the manuscript the original control and suppressor strains as well as the new strains requested by Reviewers #2 and #3. From these data it is now clear that all the strains show comparable growth behavior on galactose-containing medium. We also repeated the western blot experiments to detect HA-Dbp7 levels with a Pgk1 loading control in the Sup. #10 suppressor and control strain (4 replicates). We now show accurately in the revised manuscript that HA-Dbp7 levels do not vary in these strains under permissive conditions (galactose-containing medium), while they become undetectable under repressive conditions (glucose). These data (2 replicates) are now presented in (Supplementary Fig. 3) to take into account the suggestion of Reviewer #1.

3) Page 11 the authors indicate that loss of function of snR190 leads to defects on maturation pathway of 25S and 5.8S. This is only shown for 25 S, not for 5.8S ?

We thank Reviewer #3 for raising this concern which allowed us to improve the accuracy of the description of the phenotypes in the revised manuscript. This point is also related to minor point #2 of Reviewer #2. During the revision period, we tested the accumulation levels of the 7S precursors and mature 5.8S rRNA in absence of snR190 and we did not see any significant

decrease. Our statement in the original manuscript was therefore not accurate and we apologize for this approximation. We suspect that the 27SA₂ to 27SB conversion defect in absence of snR190 is not severe enough to impair 7S precursor production. The Northern blot data showing the accumulation levels of the 7S pre-rRNA and mature 5.8S rRNA are now presented in **Supplementary Fig. 6 (7S)** and **Supplementary Fig. 7 (5.8S)** of the revised manuscript and we have modified our statement accordingly (**page 8, line 20**).

4) Figure 5C: can the authors comment on snR42 which is also displaced in dbp7 mutant ? on snR5 large decrease in dbp7 mutant ?

We understand this comment of Reviewer #3. We would like to stress that the only reliable information that can be drawn from these sedimentation profiles are intra-series quantifications, such as the pre-ribosome-bound *versus* free ratios presented in **Supplementary Fig. 10** and we only drew conclusions from these ratios. We wrote a note in the legend of **Fig. 5c** to explain this (**page 36, line 17**). We agree that snR42 is displaced in the *dbp7Δ* mutant strain, but to a lesser extent compared to snR190 and snR37, and moreover in the opposite direction compared to these latter snoRNAs, namely the proportion of free snR42 snoRNPs increases in the *dbp7Δ* strain.

Concerning the decrease in snR5 levels in the mutant, this is actually not a biological effect but a visual artifact due to the mounting of the figure with an underexposed Northern blot for the *dbp7Δ* strain. In the revised version of the manuscript, we chose a longer exposure of the snR5 signals for the *dbp7Δ* strain, bearing in mind that what we are analyzing are the relative amounts of the bound *versus* free fractions of a given snoRNA and how this ratio differs in the wild-type and *dbp7Δ* strains.

5) Was it shown that the Dbp7 K197A ATPase activity is null ?

We must acknowledge that we performed the *in vivo* studies assuming that the mutation was indeed affecting the catalytic activity of Dbp7. The behaviour of the strain in terms of growth and ribosomal subunit profiles on gradients were consistent with a significant inactivation of Dbp7 but we did not show it formally. To address this concern, we tested the activity of the wild-type and Dbp7-[K197A] mutant using *in vitro* NADH-coupled ATPase assays. These data now show unambiguously that the K197A mutation strongly affects the ATPase activity of Dbp7. This figure has been added in the revised manuscript (**Fig. 6a, b**) and described in the main text (**page 12, line 28**).

6) Fig 6A: The lack of snR190 alleviates the growth defect of a dbp7 null mutant, is that true also for the dbp7 K197A mutant ?

To address this question, we transformed the *dbp7Δ* strain and the *snR190-[mut.C] dbp7Δ* double mutant strain with expression vectors expressing wild-type Dbp7 or Dbp7-[K197A] mutant and performed serial dilution growth assays. We observed that expression of Dbp7-[K197A] partially rescues the growth defect of the *dbp7Δ* strain, indicating that this mutant version of Dbp7 retains some levels of functionality. Importantly, as asked by Reviewer #3, snR190 loss-of-function also partially alleviates the growth defect of the *dbp7Δ* strain expressing Dbp7-[K197A]. These data are presented in **Supplementary Fig. 12** of the revised manuscript and described in the main result section (**page 13, line 28**).

7) The pre-rRNA processing defects observed in the absence of snR190 are present but weak. Authors should be careful since differences are observed between different clones (see Figure

2A, level of 25S in WT 1 and 2, level of 25S between snr190 mut C 1 and 2; also in Supp Fig3, in Fig 4B, 27SA2 and 27SB in mutAB 1 and 2 are very different). The authors should define more clearly the error bars. 2 clones were used but how many experiments were done with each?

We agree with Reviewer #3 on the variations among clones. For the processing defects in absence of snR190, we have worked with two independent clones bearing chromosomal snR190 box C mutations in both the BY4741 and W303 backgrounds (Fig. 2a and Supplementary Fig. 4a). This constitutes four biological replicates and the Northern blotting experiments have been performed at least twice with all the strains, which constitutes at least two technical replicates. The impact of snR190 loss-of-function on pre-rRNA processing was also independently assessed in Fig. 2b (W303), Fig. 4b (W303) and Supplementary Fig. 8b (BY4741) where the snR190-[mut.C] strains generated using the CRISPR-Cas9 approach were transformed with the empty vector. We showed an additional replicate in the inputs of the IPs presented in Fig. 5b. During the revision period, we performed an additional replicate to show in parallel the processing defects in strains *snr190-[mut.C]* and *dbp7Δ* (Supplementary Fig. 6a). In all these cases, we observed consistently a substantial accumulation of the 35S, 33S/32S, a slight accumulation of the 27SA₂ intermediates and a decrease in 27SB levels. We have quantified again these changes and the 27SAB/27SA₂ ratios including as many replicates as possible, and we now provide accurate quantifications and statistical analyses when possible (Supplementary Figs. 5, 6, 7, 8). We precisely mentioned in the corresponding figure legends the number of replicates studied and the results of the statistical analyses.

Concerning the impact of mutations of boxes A, B and A+B of snR190, we have worked with two independent clones in both the BY4741 and W303 backgrounds (four biological replicates) analysed as one technical replicate for the W303 background or two technical replicates for the BY4741 background. Bar charts in Supplementary Fig. 8a, c show the average values for each strain background and the error bars represent the standard deviations. A statistical analysis has been added for the BY4741 background out of the 4 replicates (Supplementary Fig. 8c).

8) Fig 3B: a WT strain could have been shown on the same figure

We understand the point that a wild-type strain could have been added in this experiment to detect the methylation levels in a “fully wild-type” strain. However, we did not judge this absolutely required, because we felt that the experiments presented in our original manuscript were already appropriately controlled. We needed to test the [mut.S] version of snR190 to challenge our hypothesis that the internal stem was potentially preventing methylation to be synthesized. We chose to generate the mutation on the *SNR190-U14* expression plasmid, not on the chromosome using the CRISPR-Cas9 technique. In this system, we reasoned that the accurate controls were the *snR190-[mut.C]* strain transformed either with the plasmid expressing wild type *SNR190-U14* or with the empty vector. We hope Reviewer #3 will understand and accept this argument.

Minor points, typos

1) Fig 2A: What is the difference between strains W303 and WT1, WT2.

We agree with Reviewer #3 that these strains needed a more precise description. The strains referred to as “WT” in our study correspond to cells that have undergone independently the CRISPR-Cas9 mutagenesis process but which turned out to lack mutation in snR190 box C at the end of the screening. We were worried about off-target mutations introduced by the CRISPR-

Cas9 procedure. Indeed, expression of the Cas9 in the cells and then growth on 5-fluoroorotate to counter-select the Cas9-encoded plasmid may have been a source of random mutagenesis. We therefore felt that these strains constitute perfect controls since they have been handled exactly as the mutant clones but express a wild-type version of snR190. Since we studied in depth two independent clones in the two different background (BY4741 and W303), we are pretty confident that the phenotypes are reliable. We have clarified this point in the revised version of the manuscript (Page 7, line 21).

2) Supp Table 2 : there is a difference between doubling time of W303 snR190mutC and the same strain with empty vector which renders the interpretation of complementation difficult ? This is not observed in BY4741 background.

We thank Reviewers #3 and #2 (Minor point #3) for having noticed this inconsistency that we missed in the original submission. We repeated the experiments with all the strains in the W303 background and calculated the corresponding doubling times. For reasons that we retrospectively do not understand, the doubling time of the W303 *snR190-[mut.C]* with the empty vector was not correct. We now provide the accurate values in **Supplementary Table 2** and **Supplementary Fig. 15** of the revised manuscript.

3) Supp Fig 2: the pink color used both for snoRNA and 25S brings confusion.

We agree with this comment of Reviewer #3. The colours were actually different but the tints were too close. We modified the figure in the revised manuscript by choosing more contrasted colours.

4) Supp Fig 4B row 7 and 8: write mutB instead of mutA.

We thank Reviewer #3 for having noticed this mistake, which was corrected in the revised manuscript.

5) Results p7: the genetic link is shown between Dbp7 and 25S rRNA, not directly with snR190. Reviewer #3 is completely right. We changed the title of the paragraph to "*Genetic link between Dbp7 and snR190 base-pairing site on 25S rRNA*" in the revised manuscript (page 6, line 2).

6) Figure 6A: position of pre60S and 90S should be indicated.

We assume that Reviewer #3 was referring to Supplementary Fig. 6 instead of Fig. 6, since the later does not contain sucrose gradient fractionations. Concerning Supplementary Fig. 6, it is not easy to answer this comment because the data result from acrylamide Northern blotting experiments suited for the detection of small RNA molecules but not of the large precursors contained in 90S particles for example. From other similar experiments (but rigorously not in this one), we know that the small 7S precursor to the 5.8S rRNA, which is present in pre-60S particles, overlaps approximately with the 60S peak. From other experiments, 35S pre-rRNA (90S particles) smears from the 60S peak to heavier fractions. To address this comment as much as we could, we labelled the position of the (pre-)60S and 90S particles on the figure (now **Supplementary Fig. 11** of the revised manuscript) and we mentioned in the corresponding figure legend that these positions were approximate, based on previous studies with relevant references (page 5, line 26 of the revised Supplementary Figure and Table legends).

REVIEWER COMMENTS

Reviewer #2 (Remarks to the Author):

In the revised manuscript the authors have sufficiently dealt with all my raised concerns and further improved manuscript. The paper should now be published.

Reviewer #3 (Remarks to the Author):

Jaafar et al present a fully revised manuscript, taking into account my comments and those from the two other referees. New data were included making this manuscript suitable for publication.

Two minor points:

- concerning my previous minor point 3, I still think that colors could be more contrasted between snoRNA and 25S.
- Figure 1 have been considerably improved following remarks from the three referees. Do the authors have the plates from 1b incubated 4 days as in 1a ? or just with the same background ?
Could the indications on each line of Fig1a be more explicite (for example, Gal1::DBP7, Gal7::35S, Poll-35S, Poll35S C2392T...?)

REVIEWERS' COMMENTS

Reviewer #2 (Remarks to the Author):

In the revised manuscript the authors have sufficiently dealt with all my raised concerns and further improved manuscript. The paper should now be published.

We thank Reviewer #2 for this very positive evaluation.

Reviewer #3 (Remarks to the Author):

Jaafar et al present a fully revised manuscript, taking into account my comments and those from the two other referees. New data were included making this manuscript suitable for publication. Two minor points:

- concerning my previous minor point 3, I still think that colors could be more contrasted between snoRNA and 25S.

We changed the colors in Supplementary Fig. 2. We now use green for the snoRNA antisense elements and pink for the 25S rRNA complementary sequences, which should be sufficiently contrasted. To remove any further ambiguity, we also changed the color of the mutations introduced in snR190 snoRNA (panels b and c) from pink to red. The corresponding figure legend has been updated accordingly.

- Figure 1 have been considerably improved following remarks from the three referees. Do the authors have the plates from 1b incubated 4 days as in 1a ? or just with the same background?

We replaced the image of Fig. 1 panel b with the image of the same plate incubated for four days instead of three. We updated the corresponding figure legend accordingly.

Could the indications on each line of Fig1a be more explicite (for example, Gal1::DBP7, Gal7::35S, Poll-35S, Poll35S C2392T...)?

We changed the annotations in Fig. 1a according to the suggestions of Reviewer #3.